# Refining Prognostic Factors in Adult-Onset Multiple Sclerosis: A Narrative Review of Current Insights

**DOI:** 10.3390/ijms26167756

**Published:** 2025-08-11

**Authors:** Tommaso Guerra, Massimiliano Copetti, Mariaclara Achille, Caterina Ferri, Marta Simone, Sandra D’Alfonso, Maura Pugliatti, Pietro Iaffaldano

**Affiliations:** 1Department of Translational Biomedicine and Neurosciences-DiBraiN, University of Bari “Aldo Moro”, Azienda Ospedaliero-Universitaria Consorziale Policlinico di Bari, 70100 Bari, Italy; guerra.tommaso93@gmail.com (T.G.); mariaclarachille@gmail.com (M.A.); 2Unit of Biostatistics, Fondazione IRCCS “Casa Sollievo della Sofferenza”, 71013 San Giovanni Rotondo, Italy; m.copetti@operapadrepio.it; 3Child Neuropsychiatry Unit, Department of Precision and Regenerative Medicine, Jonic Area University of Bari “Aldo Moro”, 70100 Bari, Italy; marta.simone@uniba.it; 4Department of Neuroscience, S. Anna University Hospital, 44100 Ferrara, Italy; caterina.ferri@unife.it (C.F.); pglmra@unife.it (M.P.); 5Department of Health Sciences, University of Piemonte Orientale, 28100 Novara, Italy; sandra.dalfonso@med.uniupo.it; 6Department of Neuroscience and Rehabilitation, University of Ferrara, 44100 Ferrara, Italy

**Keywords:** multiple sclerosis, prognosis, biomarkers, progression

## Abstract

Multiple sclerosis (MS) is characterized by a continuum of diverse neuroinflammatory and neurodegenerative processes that contribute to disease progression from the earliest stages. This leads to a highly heterogeneous clinical course, requiring early and accurate prognostic assessment: the identification of reliable prognostic biomarkers is crucial to support therapeutic decision-making and guide personalized disease management. In this narrative review, we critically examined the current MS literature, investigating prognostic factors associated with disease progression and irreversible disability in adult-onset MS, with a focus on different clinical, radiological, and molecular biomarkers. Particular attention is directed toward the prognostic value of baseline clinical and neuroimaging factors, emerging biomarkers of smoldering disease, and progression independent of relapse activity (PIRA) events. Additionally, we discussed the role of integrated prognostic tools and risk scores, as well as their potential impact on clinical practice. We aim to provide a comprehensive and clinically oriented synthesis of available evidence in the MS biomarkers field, supporting multifaceted prognostication strategies to improve long-term outcomes in people with MS.

## 1. Introduction

Neurologists have long pursued the identification of robust prognostic factors to anticipate the variable course of multiple sclerosis (MS), both in clinical care and research contexts [1,2,3]. Comparable to a complex maze that defies linear solutions, this challenge requires an integrated and multi-pronged approach. To enable early diagnosis and accurately identify MS patients at elevated risk of disease progression, precise diagnostic criteria and prognostic markers are both essential [4,5]. The predictive value of single clinical biomarkers remains moderate when considered alone, emphasizing the need for a comprehensive prognostic model incorporating imaging, molecular, clinical biomarkers, and genetic data. This is prodromal to the tailoring of individual treatment strategies to maximize clinical outcomes in the wide and constantly expanding field of disease-modifying therapies (DMTs) for MS [6]. According to the initial course of the disease, MS is typically categorized as either relapsing–remitting (RR) or primary progressive (PP); when disability accumulates, a secondary progressive (SP) phenotype is diagnosed [7]. Individuals with MS often experience a deterioration of their motor and cognitive performance, even when their inflammatory parameters remain stable: progression independent of relapse activity (PIRA) is now recognized as the epiphenomenon of smoldering MS disease. Since the disease’s onset and throughout its progression, PIRA and relapse-related worsening (RAW) episodes both contribute to irreversible neurological impairment [8,9]. In this *continuum*, the assessment of prognostic factors, including the newest outcomes, becomes crucial to capturing the evolution of the disease and guiding patient management and appropriate therapeutic choices. Potential predictors of MS progression have been the subject of numerous investigations [10,11,12]; however, clinical applications and widely accepted scores have been limited due to the lack of a systematic synthesis of evidence and the heterogeneity of findings [11,12]. The purpose of this study is to present an extensive and updated narrative review of evidence related to identified prognostic factors for adult-onset MS. The aim is to gather and thematically organize prognostic factors in MS, providing a synthesis that is relevant both from a clinical and research perspective.

## 2. Methods

### 2.1. Aims and Research Planning

This narrative review aims to provide a comprehensive and up-to-date overview of prognostic factors in adult-onset MS. Starting from a scrutiny of MS literature, we examined and discussed numerous topics in the different thematic sections of this review. A narrative review format was adopted to provide a broad overview of the heterogeneous literature on prognostic factors in MS, integrating evidence across various clinical, molecular, and radiological domains. Due to the wide variability in study designs, outcomes, and cohorts, the narrative approach allowed a wider interpretative perspective, essential to contextualize the clinical implications of this review.

### 2.2. Study Selection

A specific approach for selecting literature sources has been applied in the two primary academic databases, PubMed and Google Scholar. Our search strategy comprised pertinent keywords related to prognosis in MS. During the selection procedure, particular inclusion and exclusion criteria were applied to guarantee the high standard and applicability of this review. The main focus was about the prognostic role of different biomarkers in MS: studies investigating demographic, clinical, magnetic resonance imaging (MRI), laboratoristic biomarkers, and their association with various outcomes of clinical and radiological disease activity were included. Factors associated with DMTs exposure were also considered. Environmental exposures and genetic predispositions were not taken into consideration in this review.

Each identified published study was classified as high or low quality, depending on study design, population size, risk of bias, and assessment of outcomes [13]. Following this search path, some papers were eliminated since they were either unpublished manuscripts or non-peer-reviewed materials.

Specifying the time frame of the search, no restrictions were placed on publication dates, but we focused mainly on studies published in the last years. The time span of the references included in this review ranges from 2003 to 2025. Notably, the majority of the cited literature (approximately 70%) was published from 2020 onward, reflecting the most recent advancements in MS research field. All of the prognostic factors that were taken into consideration were listed in a summary table (Table 1) reporting the main studies cited in the text along with their key characteristics. Notably, we extracted the study-specific relative risk estimates (risk ratio, odds ratio [OR], hazard ratio [HR], or other measures of statistical association) together with the corresponding confidence intervals. Table 1 reports the details of the studies considered, including the year of publication, the study design, the population size, the outcomes investigated, the effect measures (with statistical specifications), the evidence of relevance to MS progression/worsening, and the reference in AMA format. Only prognostic factors with a higher strength of evidence were included and discussed in this narrative review. A flowchart detailing the selection and screening process is provided in Figure 1.

### 2.3. Role of the Funding Source

This study falls into the framework of “PROMISING study” (Next Generation EU—NRRP M6C2—Investment 2.1 Enhancement and Strengthening of Biomedical Research in the NHS—PNRR-MAD-2022-12376868), which aims to predict MS disease progression through the development of a prognostic score. Environmental and genetic prognostic factors, as well as those specifically related to pediatric-onset MS, will be thoroughly addressed in other studies that are part of the network of this project.

## 3. Demographic and Clinical Prognostic Factors

Demographic characteristics have been defined as classic predictors of disease course in MS, widely used by neurologists in clinical practice to orient treatment choices [82]. Age constitutes a multifaceted element in prognosis: age at MS onset, in the first place, may impact the disease course. Older age (>40 years) at onset was associated with a higher risk of SPMS conversion [22] and PIRA events [23,26]. This demographic factor resulted in a significant risk of SPMS, especially in men [20]. Considering disability milestones, age at onset greater than 50 years was significantly associated with a higher risk of reaching an irreversible Expanded Disability Status Scale (EDSS) 6.0 [24,27] in a shorter time [25] compared to patients with younger age at onset. Similarly, lower age at treatment initiation has been linked to an enhanced treatment effect on annualized relapse rate (ARR) and disability progression [26], as demonstrated in a meta-analysis of six randomized clinical trials [16].

The dual role of aging in MS as a prognostic factor has to be thoroughly evaluated: reduced inflammatory activity goes in parallel with an increase in the risk of irreversible disability accumulation linked to PIRA [26]. Age is relevant in determining disability progression in MS, with pediatric-onset MS characterized by a less steep increase in EDSS scores over time than older patients and a less pronounced effect of PIRA in accelerating EDSS progression [83]. Considering inflammatory activity, patient age is the most important determinant of decline in relapse incidence [14,19]. Therefore, knowing how aging phenomena impact immune and brain cell activity may help reduce non-relapses-related progression in MS patients [84].

Male sex has been linked to poor long-term outcomes in MS, according to numerous studies [82,85,86]. Specifically, male sex has been associated with a higher risk of reaching EDSS 6.0 and 7.0 [57]; conversely, female sex appeared to display a lower risk of reaching EDSS 3.0 [33] and exerted a protective role in the late-onset cohort for the risk of a 12-month confirmed disability worsening [43]. Female sex, younger age, and a higher EDSS during relapse were considered factors associated with a higher chance of EDSS improvement after relapse treatment [17]. However, some cohort studies have reported contradictory results, with no difference in prognosis between males and females [70,87].

Early clinical characteristics such as relapse frequency, recovery from relapses, and onset symptoms have also been recognized as crucial prognostic indicators [28]. A brainstem, cerebellar, or spinal cord syndrome was associated with poor recovery from the initial relapse [58], in parallel with the solid association of multifocal onset with a higher risk of SPMS [22]. The presence of motor, especially spinal, and brainstem symptoms at onset were associated with a shorter time to irreversible EDSS 6.0 [25,57], while patients presenting clinical isolated syndrome (CIS) with optic neuritis appeared to display a lower risk of reaching an EDSS score of 3.0 [33]. In addition, an incomplete recovery from the first attack was significantly associated with a higher hazard ratio to reach an EDSS of 6.0 [27,35]. A higher baseline EDSS score was associated with a higher risk of SPMS and was a predictor of EDSS worsening in numerous studies [19,22,29,30]. The early years of disease are considered crucial, both for prognosis assessment and—as discussed below—to exploit the most optimal therapeutic response. Change in EDSS from baseline to 24 months was a strong predictor of disability outcomes over 15 years [88]. A higher number of early PIRA and RAW events led to a higher risk of SPMS in a recent MSBase observational cohort study [89]. Frequent relapses in the first two years from the onset and shorter first inter-attack intervals predicted also a shorter time to reach disability endpoints EDSS 6.0, 8.0, and 10 [27,57,65]. Relapse frequency during the RRMS phase was strongly associated with a higher risk of progression [22].

This discussion ought to cover cognitive impairment as predictors of the worst outcomes. Chronic depression and cognitive dysfunction were associated with adverse long-term outcomes in MS [35]. Cognitive impairment was associated with higher odds of transitioning from a relapsing–remitting course to a progressive disease course, and, interestingly, it was also associated with a higher mortality risk [34]. An Italian 10-year retrospective longitudinal study outlined that cognitive impairment at diagnosis, with particular involvement of memory and processing speed, was associated with a more than threefold risk of reaching EDSS 4.0 and a twofold risk of SPMS conversion [90]. Cognitive impairment can occur independently of other neurological symptoms and is linked to a higher risk of future neurological disability [91].

People of all ethnicities are affected by MS, and the ethnic background must be included in the discussion about prognosis. A recent review outlined that Black, Latino/Hispanic, and South Asian individuals with MS in North America and the United Kingdom appear to have an earlier age of onset. Furthermore, compared to white MS patients, Black and Latino/Hispanic MS patients in the USA were more likely to have severe symptoms at disease onset and an earlier disability accrual [92]. Notable differences in MS-specific mortality trends by age and race/ethnicity were also highlighted, indicating an unequal burden of disease and a complex balance of environmental and social differences influencing disease variability [93].

These data underline the complexity of demographic influences and the necessity of integrating multiple clinical and biological variables when estimating prognosis (Figure 2).

## 4. Radiological Predictors

The prognosis and treatment choices of patients may be improved by considering all available conventional and advanced MRI measures [4]. In individuals with CIS and RMS, a greater number of brain T2-hyperintense white matter (WM) lesions at baseline raises the probability of disability accrual, MS progression [33,74,94], and RAW events [15]. Baseline gadolinium (Gd)-enhancing lesions were also independently associated with SPMS conversion at 15 years [32]. T1-hypointense lesions (“black holes”) primarily indicate axonal degeneration, white matter disruption, and are typically linked to irreversible clinical outcomes [95,96,97]. A more intriguing prognostic factor turned out to be lesion topography. The primary predictor of progressive disease and physical impairment in a group of MS patients with very long follow-up and uniform disease duration was cortical involvement, both in terms of lesions and atrophy [36]. Cortical lesion accrual was greater in SPMS than RMS and cortical lesion volume independently predicted EDSS changes throughout the disease course [37]. The number of spinal cord (SC) lesions on MRI was associated with future accumulation of disability largely independent of relapses [15,72]. A significant association between new SC lesions and clinical relapses within 3 months was found, turning this prognostic factor into a major driver of treatment change [61]. SC lesions also showed consistent association with EDSS and MS progression at 15 years [32]. The role of the optic nerve, recently included as the fifth topography for MS diagnostic criteria fulfillment [4] is triggering. The length of optic nerve lesions at onset correlated with the extent of retinal damage, as assessed by optical coherence tomography (OCT) parameters, and was associated with poorer visual recovery 12 months after disease onset [98].

A recent consensus established biomarkers of chronic active lesions (CAL), crucial signs of chronic inflammation: paramagnetic rim lesions (PRL) identified on susceptibility-sensitive MRI, MRI-defined slowly expanding lesions (SELs), and 18-kDa translocator protein (TSPO)-positive lesions on positron emission tomography (PET) [99]. PRLs have been linked to a more severe course of the disease [100], without a clear correlation with their topographical distribution. The number of PRL was also associated with the number of leptomeningeal contrast enhancement foci, linking leptomeninges to mechanisms related to sustaining chronic inflammation [101]. Additionally, PRLs have been connected to increased rates of atrophy in the brain and spinal cord [102]. The amount of SELs has been correlated with MS progression after 9 years, and severe SEL microstructural abnormalities were a predictor of EDSS worsening and SPMS conversion [29]. A higher definite SEL volume was associated with increasing disability, assessed by EDSS, z scores of the Multiple Sclerosis Functional Composite, Timed 25-Foot Walk Test and Paced Auditory Serial Addition Task, and increased risk of clinically defined progression [103]. In a recent study, the proportion of persisting black holes was higher in SELs compared to non-SELs, and within-patient SEL and persisting black holes volumes were positively correlated [104].

In addition to the prognostic value of MRI, neuroimaging may also aid in predicting treatment response [105]. New T2 lesions, an increase in T2 lesion volume, and Gd+ lesions on MRI are considered, isolated or combined, predictors of disability progression and treatment effectiveness [106,107,108]. It is applicable to both asymptomatic and symptomatic lesions, as demonstrated by a study of the MSBase analyzing the probability of treatment change among patients with clinically silent new MRI lesions [109]. Nowadays, it is crucial to take into account an integrated approach that supports the care of MS patients [110], given the strong predictive significance of the several radiological indicators mentioned above.

Molecular mechanisms of progression phenomena may be related to the concept that iron-positive rim lesions are characterized by the presence of iron-laden activated myeloid cells and the activation of related molecular pathways: an upregulation of the CD163–HMOX1–HAMP axis at the rims of chronic active lesions was reported, suggesting that haptoglobin-bound hemoglobin represents the key source of iron uptake and indicating a pro-inflammatory transcriptional profile [111,112]. This finding is further supported by the strong association between PRL levels and CSF concentrations of sCD163 in MS patients, whereas elevated IL10 mRNA expression was observed in perilesional myeloid cells [112]. A recent review [113] highlighted the involvement of ectopic lymphoid follicles in MS, underscoring their prognostic association with both cortical [114] and spinal cord [115] pathology. The molecular profile of inflammatory meningeal and perivascular infiltrates was defined by a high density of CXCR5^+^ cells, cytoplasmic NFATc1^+^ cells, enriched populations of CD3^+^CD27^+^ memory T cells, and CD4^+^CD69^+^ tissue-resident cells [116].

## 5. Fluid Biomarkers

Immune-related biomarkers can predict future impairment and correlate with MS severity, as well as imaging and clinical outcomes [1,2,3,4,5,6,7,8,9,10,11,12,13,14,15,16,17,19,20,22,23,24,25,26,27,28,29,30,32,33,34,35,36,37,43,57,58,61,65,70,72,74,82,83,84,85,86,87,88,89,90,91,92,93,94,95,96,97,98,99,100,101,102,103,104,105,106,107,108,109,110,111,112,113,114,115,116,117]. Cerebrospinal fluid (CSF) biomarkers are useful not only in the diagnostic process: CSF samples can also enhance the biological profiling of the disease, thereby determining its long-term prognosis [118]. Historically, CSF-specific oligoclonal band (OCB) testing has been the most generally accessible laboratory test recognized as the cornerstone of MS diagnosis [119]. The presence of CSF-specific OCBs significantly doubled the risk of attaining disability milestones EDSS 4.0 or 6.0 in a meta-analysis including data from 1918 patients [62]. CSF-OCB presence was associated with a higher risk of relapses [41], accumulation of disability [33], and SPMS conversion [63]. In addition, an increased number of cortical lesions was found in OCB-positive compared to OCB-negative patients [38]. In this recent study, OCB presence at MS onset was associated with more severe gray matter pathology and with worse physical and cognitive impairment after 10 years, stressing the link with B cell activation, lymphoid-neogenesis, and pro-inflammatory immune response in the CSF of OCB+ patients [38].

The role of neurofilament light chains (NfL) elevation as a predictor of confirmed disability worsening independent of clinical relapses has been recently documented [120]. Considering neuroinflammatory activity, NfL levels were an independent factor for the occurrence of at least one relapse during the first two years after MS diagnosis and for the occurrence of Gd+ lesions during the first 2 years from diagnosis at brain and spine MRI scans [121]. Subjects with higher serum NfL Z scores showed a greater probability of relapses, EDSS worsening, and EDA in the following years [67]. Serum GFAP concentration, in contrast to serum levels of NfL, does not usually increase during acute inflammation; instead, it indicates faster grey matter (GM) brain volume loss and may serve as a predictive biomarker for subsequent PIRA [122]. Levels of GFAP correlated also with SEL count [123].

A possible prognostic neurodegenerative biomarker of GM dysfunction was suggested to be parvalbumin levels in the CSF at the time of MS diagnosis, highlighting a correlation with physical disability, fatigue, and MRI brain volume of strategic regions related to cognitive impairment [124]. The kappa free light chain (KFLC) index has recently been recognized as a diagnostic biomarker, but its prognostic role is also relevant [125]. KFLC index was an independent risk factor for PIRA [126] and was also predictive of disease activity in the first year after diagnosis [127] and during follow-up [126,127,128]. Stressing the combined use of different biomarkers, further stratification of MS disease activity risk in OCB-positive patients was possible using the KFLC index [128]. Considering the evidence of compartmentalized inflammation, a score was recently calculated based on glial and axonal markers (CHI3L1*GFAP/NfL), known as “Glia score” and related to progressive MS [129]. A detailed molecular CSF profiling, combined with clinical and radiological assessment, could serve as a prognostic marker for aggressive MS [130]. The CSF of MS patients with higher levels of GM damage at diagnosis showed a proinflammatory pattern of elevated levels of molecules linked to sustained B-cell activity and lymphoid neogenesis, such as CXCL13, IL6, IL8, and IL10; proinflammatory cytokines, such as TNF and IFNγ; and high levels of BAFF, APRIL, LIGHT, TWEAK, sTNFR1, sCD163, MMP2, and pentraxin III [130]. While levels of TNF-α exhibited a positive correlation with post-contrast-enhancing cerebral lesions and T2 cervical SC lesions, IL-6 rates were linked with post-contrast-enhancing thoracic SC lesions and IL-15 levels negatively correlated with T2 and Gd-positive lesions in cervical SC [40]. Free-circulating mitochondrial DNA (mtDNA) levels could also play a role in prognosis. A larger T2 lesion burden and EDSS worsening were positively connected with higher CSF quantities of mtDNA copies in progressive MS patients [131]. Moreover, CSF lactate levels have been connected to increased neurological impairment, and molecular biomarkers of neurodegeneration [132].

## 6. Therapies and Prognosis

DMTs have significantly improved the natural history of the disease, modifying its prognosis [1,2,3,4,5,6]. The assessment of therapeutic prognostic value is developed through four distinct dimensions of analysis: time from disease onset/diagnosis to treatment initiation, total exposure, type of treatment, and discontinuation. A delayed DMT initiation was associated with a higher risk of PIRA and RAW events in a cohort of adult-onset and pediatric-onset MS patients [26]. The time interval between disease onset and the first DMT start was a strong predictor of disability accumulation, independent of relapse activity, over the long term; in addition, an increased risk of disability accumulation was underlined for patients who started the treatment after 1.2 years from the onset [74]. A recent consensus has combined all references in favor of early intervention with high-efficacy disease-modifying therapies (HE-DMTs), representing the best window of opportunity to delay irreversible CNS damage and MS-related disability progression [133]. HE-DMTs commenced within 2 years of disease onset were associated with less disability after 6–10 years than when commenced later in the disease course [75]. Compared to patients who began treatment with DMT earlier, those who started later achieved an EDSS score of 6 faster, and their mortality rate was 38% higher [76]. Better patient-reported physical symptoms were also reported [77]. The Early Intensive Therapy (EIT) strategy was more effective than the escalation (ESC) strategy in controlling disability progression over time [78]. EIT strategy refers to the approach whereby patients are treated with HE-DMTs, including alemtuzumab, cladribine, fingolimod, natalizumab, ocrelizumab, ozanimod, and ponesimod as first therapy. Conversely, the ESC group included patients initially treated with ME-DMTs (azathioprine, interferon-beta products, glatiramer acetate, teriflunomide, dimethyl fumarate) and then escalated to HE-DMTs [133]. The risk of reaching EDSS 4.0 was reduced by 26% in patients starting with HE DMTs, in parallel with a reduced risk of relapses by 66% and a three times higher probability of confirmed disability improvement [79]. A lower probability of a first relapse and 6-month confirmed EDSS score worsening was found in patients starting an HE DMT as first therapy, compared to subjects starting moderate-efficacy DMTs (ME DMTs) [81]. Therefore, the median time to sustained accumulation of disability was longer for the EIT group [80]. A shorter DMT exposure is associated with a higher risk of PIRA event [26] and a higher risk of SPMS conversion [74]. This concept is reinforced by another study, which highlights that the occurrence of RAW events was predicted by the temporary or permanent discontinuation of the initial DMT [15]. The mechanisms linked to “silent progression” may be biologically impacted by the various modes of action of DMT utilized in clinical practice. The use of DMTs has been demonstrated to be crucial in reducing PIRA occurrence in numerous recent studies, significantly impacting prognosis [134,135,136].

## 7. Other Biomarkers with a Prognostic Value

Progressive neuronal and axonal loss is thought to be one of the primary mechanisms sustaining MS-related impairment, and this neurodegenerative process frequently involves the visual system. Consequently, orbital ultrasonography [59] and optical coherence tomography [137] became valuable non-invasive tools in the field of MS biomarkers. Combined macular ganglion cell and inner plexiform layers (mGCIPL) atrophy correlated with brain atrophy [137], and a significant thinning of the mGCIP was observed also in MS patients without a history of optic neuritis, highlighting a subclinical optic nerve involvement [138]. According to a recent review, cross-sectional measurement of peripapillary retinal nerve fiber layer (pRNFL) and mGCIPL thickness (≤88 µm and <77 µm, respectively) and longitudinal measurement of pRNFL thinning and mGCIPL thinning (1.5 µm/year and ≥1.0 µm/year, respectively) were associated with an increased risk of disability progression in subsequent years [139]. Optic nerve diameter (OND) in ultrasonography and RNFL thickness were significantly lower in patients with an EDSS score > 2 than in those with a score ≤ 2, indicating that OND was an independent predictor of EDSS > 2 [60]. Ultrasound findings and disease progression showed a substantial correlation, although there were no statistically significant changes related to relapses or other clinical factors [59].

## 8. The Strategic Role of Prognostic Algorithms in Clinical Decision-Making and Research

Combining various biomarker and treatment response measures with demographic and clinical prognostic factors has a more significant clinical impact on long-term prognosis than considering factors individually. The demonstration is the application of the Risk of Ambulatory Disability (RoAD) score, built on demographic, clinical baseline factors, and 1-year assessment of treatment response combined [31,140]. A common indicator of treatment failure is the occurrence of new T2 lesions on serial MRI. Particular thresholds in assessing lesion burden are linked to the progression of disability over time: the Canadian MS Working Group Treatment Optimization Recommendations and the modified Rio score are two examples of treatment algorithms that are utilized in clinical practice to support the clinical decision-making process [141,142,143]. These grading systems also consider both MRI results and clinical characteristics. In patients treated with teriflunomide, the MAGNIMS score predicted a 7-year probability of disability worsening, while in individuals treated with interferon beta-1a, it predicted long-term disability progression for up to 15 years [144,145]. The Multiple Sclerosis Treatment Decision Score (MS-TDS) is another example of a prediction model to assist with treatment decision-making. Combining different prognostic factors, it can identify patients who benefit from early platform medicine by estimating tailored therapy success probabilities [146]. No evidence of disease activity (NEDA) has been considered in recent years a therapeutic goal and measure of individual treatment response: in particular, NEDA-3 status requires the absence of relapses, EDSS progression, and inflammatory MRI activity, and NEDA-4 expanded this definition by adding the absence of increased brain atrophy [147,148]. A recent metanalysis highlighted that NEDA-3 and NEDA-4 at 1–2 years were associated with two times higher odds of no long-term disability progression at 6 years [149]. Considering all algorithms and scores proposed, MRI has the role of a fundamental prognostic factor for the monitoring of MS disease and treatment [150]. Using a machine learning technique, algorithms have recently been developed to predict confirmed disability accumulation, NEDA status, immunotherapy initiation, and the escalation from low- to high-efficacy therapy with intermediate to high accuracy [151,152,153].

A very recent prognostic tool is the Barcelona Risk Score (BRS), a validated algorithm that incorporates several biomarkers to classify each patient into four data-driven groups according to the risk of moderate long-term disability, considering different outcomes: RAW, PIRA, SPMS conversion, MRI features, and patient-reported scores. The BRS offers a versatile framework, designed to support clinical decision-making in everyday practice and across heterogeneous settings, and it is applicable even with limited data availability [154].

In conclusion, our practical recommendation is to adopt integrative algorithms capable of a comprehensive assessment of patients, by combining biomarkers that reflect the disease from multiple angles, ranging from clinical presentation [155] to molecular indicators of progression, and neuroimaging findings.

## 9. Conclusions

This review outlines key research findings regarding the prognostic role of diverse biomarkers in MS. Some limitations need to be acknowledged. As a narrative review, this study lacks the methodological rigor of a systematic review and is therefore potentially subject to selection bias and other inherent limitations related to the non-systematic inclusion of studies. Generalizability may also be impacted by variations in study designs, populations, and prognostic factor definitions. Furthermore, the capacity to assess prognostic importance or provide pooled estimates is restricted by the lack of quantitative synthesis. Additionally, it should be noted that many of the prognostic factors addressed in this review are not mutually independent: this reflects the intrinsic complexity of prognostication in MS, where different biomarkers often overlap and interact. Considering strengths and limitations of this narrative review, an up-to-date overview of the long-term prognostic value of different biomarkers could aid clinicians in better considering certain aspects of clinical practice, starting from demographic features to MRI and molecular biomarkers. Expanding disease registries to incorporate as many biomarkers of disease progression as possible would promote the idea of merging datasets to provide a multifaceted picture of MS patients [156]. In this perspective, our work could provide a comprehensive overview of prognostic factors in MS research in an integrated way, constituting a roadmap for future researchers in their efforts to contribute to existing studies aimed at improving MS care.

## Figures and Tables

**Figure 1 ijms-26-07756-f001:**
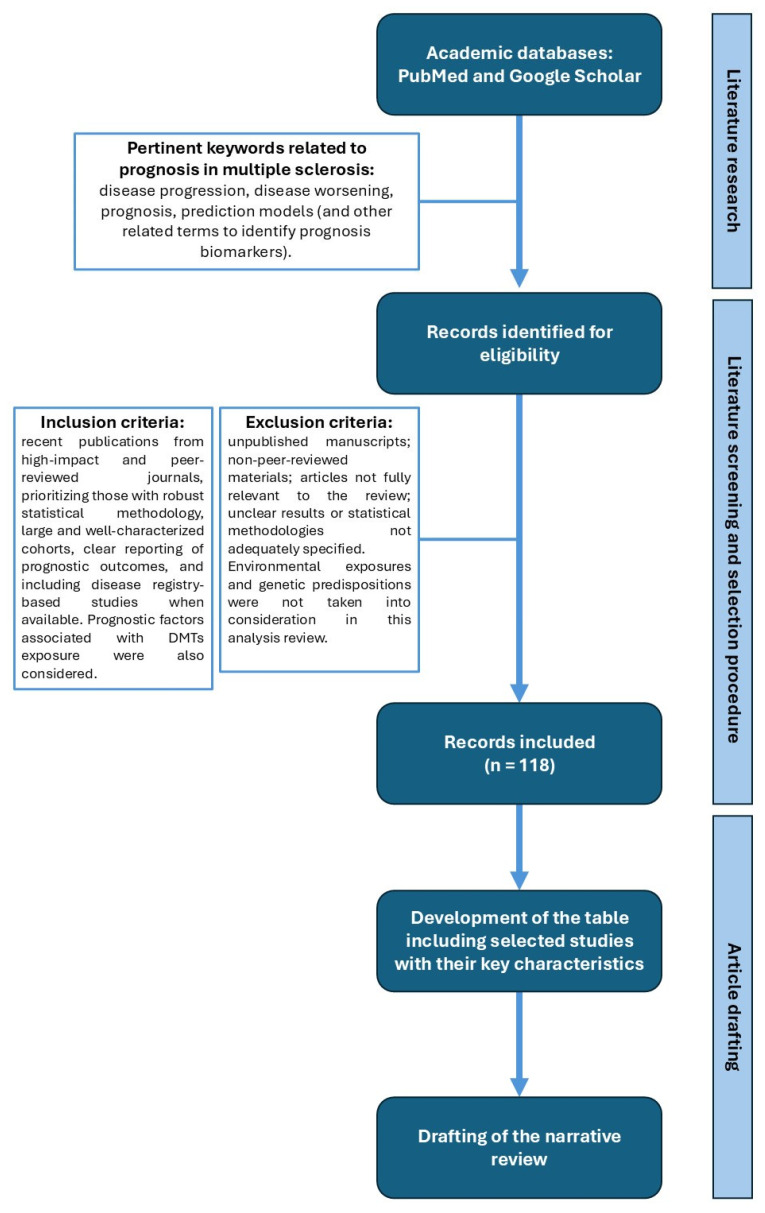
Flowchart of the methodology for identification, screening, and inclusion of studies in this narrative review.

**Figure 2 ijms-26-07756-f002:**
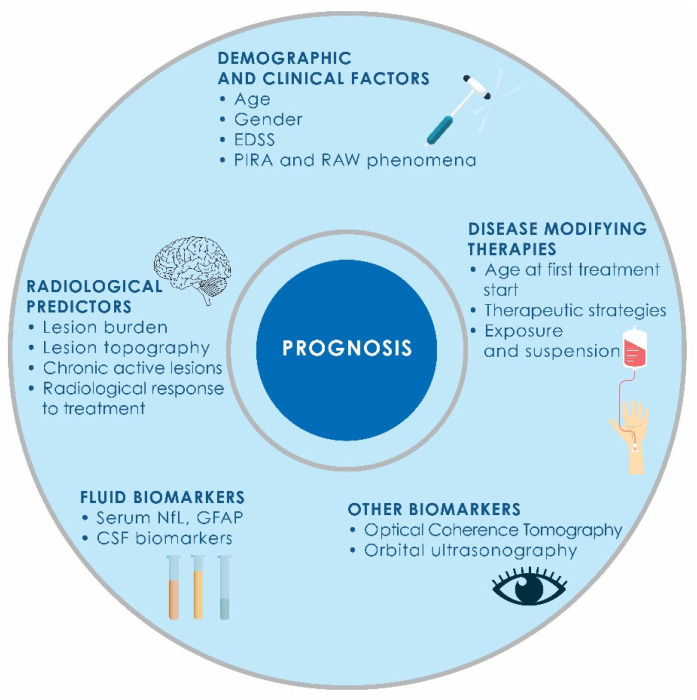
Graphical overview of prognostic factors in multiple sclerosis.

**Table 1 ijms-26-07756-t001:** Overview of the literature cited in the text and reviewed as part of the research for this narrative review, focusing on key clinical, radiological, and molecular prognostic factors in multiple sclerosis.

	Prognostic Factor	Year of Publication	Study Design	Population Size	Outcome(s)	Effect Measure	**Evidence of Relevance to MS Progression/Worsening**	**References**
**Demographic and clinical prognostic factors**	**Age**	2013	Retrospective multi-center cohort study	12,570 relapse-onset patients and 881 patients with PPMS	Relapse incidence	HR = 0.95, 95% CI = 0.949–0.953, *p* < 10^−12^. Tweedie model: rate ratio = 0.98, 95% CI = 0.982–0.985, *p* < 10^−12^	Patient age is the most important determinant of decline in relapse incidence.	[14]
**Demographic and clinical prognostic factors**	**Age**	2021	Retrospective 2-center cohort study	687 RRMS patients	PIRA	Sub distribution HR = 1.05 (1.01–1.10) for each year increase, *p* = 0.036	The risk of PIRA was associated with increasing age.	[15]
**Demographic and clinical prognostic factors**	**Age**	2015	Metanalysis: 6 trials	6693 RRMS	Treatment effectiveness: ARR; disability progression (EDSS worsening sustained for 12 or 24 weeks). Relative effect (RE)	Treatment effects on ARR (RE = 0.83 vs. RE = 1.30, *p* < 0.001) and on disability progression (RE = 0.82 vs. RE = 1.28, *p* = 0.017) were significantly higher in younger subjects.	In RRMS, lower age is associated with higher treatment effects.	[16]
**Demographic and clinical prognostic factors**	**Age**	2024	Retrospective cohort	114 RMS	Change in EDSS from first assessment at relapse to EDSS after the last relapse treatment	Regression coefficient (95% CI): 0.04 (0.02, 0.06) *p* < 0.001	Female sex, younger age, and a higher EDSS during relapse as factors associated with a higher chance of EDSS improvement after relapse treatment.	[17]
**Demographic and clinical prognostic factors**	**Age**	2019	Retrospective cohort	2083 RRMS	Global disability (eight Performance Scales (PSS-8) and the PHQ-9)	PSS-8: 0.65 (0.49, 0.82) < 0.001; PHQ-9: 0.39 (−0.58, −0.19) < 0.001	Older age is associated with higher global disability.	[18]
**Demographic and clinical prognostic factors**	**Age**	2018	Retrospective multi-center cohort study	4842 MS patients	Risk of relapses after DMT discontinuation	HR (95% CI) *p*-value: 0.97 (0.97, 0.98) < 0.001	In younger patients the risk of relapses after DMT suspension is higher.	[19]
**Demographic and clinical prognostic factors**	**Age at onset**	2015	Prospective study	305 MS patients	SPMS conversion	HR: 1.049; *p* = 0.00426	The factor “age at onset” was significant for risk of SP in men.	[20]
**Demographic and clinical prognostic factors**	**Age at onset**	2015	Prospective study	305 MS patients	Death (EDSS 10)	HR: 1.061; *p* = 0.0135	In men, age at onset remained a significant predictor of EDSS10.	[20]
**Demographic and clinical prognostic factors**	**Age at onset**	2022	Retrospective cohort study	661 MS patients	EDSS worsening	95% CI: 0.04 to 0.40; *p* = 0.015	For every 5 years earlier, the EDSS was 0.22 points worse.	[21]
**Demographic and clinical prognostic factors**	**Age at onset**	2022	Retrospective cohort study	661 MS patients	SPMS conversion	95% CI: 1.08 to 1.64; *p* = 0.008	For every 5 years earlier, odds of SPMS 1.33 times higher.	[21]
**Demographic and clinical prognostic factors**	**Age at onset**	2022	Retrospective cohort study	661 MS patients	Brain T2-lesion volume (T2LV)	95% CI: 1.02 to 2.70; *p* < 0.001	For every 5 years earlier, odds of T2LV 1.86 mL higher.	[21]
**Demographic and clinical prognostic factors**	**Age at onset**	2020	Retrospective multicenter cohort study	19,318 RRMS; 2343 SPMS identified with the DDA definition and 3868 identified with the neurologist definition (ND)	SPMS conversion (SPMS definition according to ND and DDA)	DDA group: HR (95% CI): 2.26 (1.92–2.67), *p* < 0.0001; ND group: 1.85 (1.63–2.09), *p* < 0.0001	Age at onset > 40 years is associated with higher risk of SPMS	[22]
**Demographic and clinical prognostic factors**	**Age at onset**	2023	Retrospective analysis of data from patients prospectively included (patients with a first demyelinating attack)	1128 patients	PIRA	HR, 1.43; 95% CI, 1.23–1.65; *p* < 0.001 for each older decade	Older age at the first attack is a predictor of PIRA.	[23]
**Demographic and clinical prognostic factors**	**Age at onset**	2017	Monocentric retrospective study	3597 pMS	Time to EDSS 4.0 and 6.0	HR 2.0 [95% CI 1.7–2.4] and 2.3 [1.9–2.9]	Worst outcomes with LOMS (≥50 years) (independent of PP course or male gender).	[24]
**Demographic and clinical prognostic factors**	**Age at onset**	2020	Retrospective monocentric study	157 PPMS patients	Time to EDSS 6.0	HR (95% CI): 1.03 (1.006–1.053); *p* = 0.012	Older age of onset was associated with a shorter time to EDSS6	[25]
**Demographic and clinical prognostic factors**	**Age at onset**	2023	Retrospective analysis of data from patients prospectively included (patients with a first demyelinating attack)	1128 patients	Adjusted yearly EDSS increase rates	HR 0.18; 95% CI, 0.16–0.20 vs. 0.04; 95% CI, 0.02–0.05; *p* < 0.001	Older age at the first attack is a predictor of PIRA.	[23]
**Demographic and clinical prognostic factors**	**Age at onset**	2023	Retrospective multicenter cohort study	16,130 MS patients	PIRA	AOMS vs. POMS HR, 1.42; 95% CI, 1.30–1.55; LOMS vs. POMS HR, 2.98; 95% CI, 2.60–3.41; *p* < 0.001.	Older age at onset was associated with a higher risk of PIRA events.	[26]
**Demographic and clinical prognostic factors**	**Age at onset**	2020	Prospective study	415 MS patients	Risk of EDSS 6.0	HR 3.846, 95% CI 1.240–11.932, *p* = 0.020	Age at disease onset greater than 50 years was significantly associated with a higher HR to reach an EDSS of 6.0.	[27]
**Demographic and clinical prognostic factors**	**Baseline EDSS score**	2020	Systematic review (30 studies). Data collection was guided by the checklist CHARMS and PROBAST.	N/A	N/A	N/A	The single most common clinical predictor was baseline EDSS (n = 11).	[28]
**Demographic and clinical prognostic factors**	**Baseline EDSS score**	2020	Retrospective multicenter cohort study (RISM)	19,318 RRMS; 2343 SPMS identified with the DDA definition and 3868 identified with the neurologist definition (ND)	SPMS conversion (SPMS definition according to ND and DDA)	DDA group: HR (95% CI) 1.41 (1.38–1.44), *p* < 0.0001; ND group: 1.50 (1.48–1.53), *p* < 0.0001	A higher baseline EDSS score is associated with higher risk of SPMS	[22]
**Demographic and clinical prognostic factors**	**Baseline EDSS score**	2022	Prospective, longitudinal cohort study	53 RRMS	EDSS score increase of ≥1.5, 1.0, or 0.5, confirmed after a 3-month relapse-free period, when the baseline EDSS score was 0, ≤5.5, or ≥6.0, respectively	A higher baseline EDSS score (OR = 3.15 [95% CI = 1.61; 8.38], *p* = 0.003) is a significant independent predictor of EDSS score worsening at follow-up (C-index = 0.892)	A higher baseline EDSS score is a predictor of EDSS worsening.	[29]
**Demographic and clinical prognostic factors**	**Baseline EDSS score**	2022	Prospective, longitudinal cohort study	53 RRMS	SPMS conversion	A higher baseline EDSS score (for each point higher: OR = 6.37 [1.98; 20.53], *p* = 0.002) independently predicted SPMS conversion (C-index = 0.947).	A higher baseline EDSS score is predictor of SPMS conversion	[29]
**Demographic and clinical prognostic factors**	**Baseline EDSS score**	2016	Post hoc analysis PRISMS long-term follow-up	382 patients	Risk of EDSS 6.0 and time to EDSS 6.0	R2 1.4125, 1.0862	There is an association between EDSS at baseline and EDSS 6.0 and time to EDSS 6.0.	[30]
**Demographic and clinical prognostic factors**	**Baseline EDSS score**	2016	Post hoc analysis PRISMS long-term follow-up	382 patients	SPMS conversion and time to SPMS	R2 0.8634; 0.6477	There is an association between EDSS at baseline and SPMS conversion and time to SPMS conversion	[30]
**Demographic and clinical prognostic factors**	**Baseline EDSS score**	2018	Retrospective multi-center cohort study	4842 MS patients	CDP	HR (95% CI) *p*-value: EDSS 2–3.5 1.79 (1.47, 2.17) < 0.001; EDSS 4.0–5.5 2.20 (1.77, 2.75) < 0.001; EDSS 6 + 2.62 (2.09, 3.28) < 0.001	Hazard of CDP increased with increasing disability at baseline.	[19]
**Demographic and clinical prognostic factors**	**Baseline EDSS score**	2021	Observational cohort study	2649 MS patients	MSIS physical score and psychological score worsening	Each year of treatment delay was associated with a worse MSIS physical score by 2.75 points (95% CI 1.29 to 4.20), and worse MSIS psychological score by 2.02 points (95% CI 0.03 to 3.78)	Earlier commencement of DMT was associated with better patient-reported physical symptoms.	[31]
**Radiological predictors**	**Baseline gadolinium-enhancing lesions**	2019	Prospective study	180 MS patients	EDSS correlation	(≥1) β = 1.32, *p* < 0.01; (≥2) OR: 3.16, 1.08, 9.23; *p* = 0.035	Baseline gadolinium-enhancing showed a consistent association with Expanded Disability Status Scale at 15 years.	[32]
**Radiological predictors**	**Baseline gadolinium-enhancing lesions**	2021	Retrospective 2-center cohort study	687 RRMS patients	RAW	Sub distribution HR = 2.38 (1.01–5.63), *p* = 0.047	RAW was predicted by the presence of contrast-enhancing lesions on baseline MRI	[15]
**Radiological predictors**	**Brain T2 lesions at baseline MRI**	2015	Observational study based on a prospective, open cohort	1018 CIS	Risk of reaching EDSS score of at least 3.0 in 2 evaluations (defined “disability accumulation”)	Adjusted HR scores of 2.9 (95% CI 1.4–6.0)	The presence ≥10 brain T2 lesions on the baseline MRI was associated with a higher risk of the accumulation of disability	[33]
**Radiological predictors**	**Brain T2 lesions at baseline MRI**	2021	Retrospective 2-center cohort study	687 RRMS patients	RAW	sHR = 3.92 (1.36–11.29), *p* = 0.012	RAW was predicted by the presence of >9 T2 lesions on baseline MRI	[15]
**Radiological predictors**	**CALs**	2022	Prospective, longitudinal cohort study	54 RRMS	EDSS score increase of ≥1.5, 1.0, or 0.5, confirmed after a 3-month relapse-free period, when the baseline EDSS score was 0, ≤5.5, or ≥6.0, respectively	A lower baseline MTR values of SELs (for each % higher: OR = 0.66 [0.41; 0.92], *p* = 0.033) is a significant independent predictor of EDSS score worsening at follow-up (C-index = 0.892)	Lower baseline MTR values of SELs are predictor of EDSS worsening.	[29]
**Radiological predictors**	**CALs**	2022	Prospective, longitudinal cohort study	56 RRMS	SPMS conversion	A lower baseline MTR values of SELs (for each % higher: OR = 0.48 [0.25; 0.89], *p* = 0.02) independently predicted SPMS conversion (C-index = 0.947).	A lower baseline MTR values of SELs are predictor of SPMS conversion	[29]
**Radiological predictors**	**CALs**	2022	Prospective, longitudinal cohort study	52 RRMS	EDSS score increase of ≥1.5, 1.0, or 0.5, confirmed after a 3-month relapse-free period, when the baseline EDSS score was 0, ≤5.5, or ≥6.0, respectively	A higher proportion of SELs among baseline lesions (OR = 1.22 [95% CI = 1.04; 1.58], *p* = 0.04) is a significant independent predictor of EDSS score worsening at follow-up (C-index = 0.892)	A higher proportion of SELs among baseline lesions is a predictor of EDSS worsening.	[29]
**Demographic and clinical prognostic factors**	**Cognitive disfunction**	2022	Retrospective study	408 MS patients	SPMS conversion	OR = 2.29, *p* = 0.043	Cognitive dysfunction was associated with higher odds of transitioning from relapsing–remitting course to a progressive disease course	[34]
**Demographic and clinical prognostic factors**	**Cognitive disfunction**	2022	Retrospective study	409 MS patients	Mortality	aHR = 3.07, *p* = 0.006	Cognitive dysfunction was associated with higher hazard of death in the total sample	[34]
**Demographic and clinical prognostic factors**	**Cognitive disfunction**	2016	Prospective observational study	793 MS patients	Reaching severe disability: EDSS 6.0 and higher	HR 4.64; CI 1.11–19.50; *p* = 0.036	Cognitive dysfunction 10 years after disease onset was associated with severe disability.	[35]
**Radiological predictors**	**Cortical lesions**	2021	Prospective study	63 MS patients	EDSS	0.37 (0.23 to 0.508); cortical lesions were higher in SPMS (100% sensitivity and 88% specificity)	Cortical lesions, grey matter volume and cervical cord volume explained 60% of the variance of EDSS; cortical lesions alone explained 43%.	[36]
**Radiological predictors**	**Cortical lesions**	2022	Prospective study	20 RRMS patients, 13 SPMS patients, along with 10 age-matched healthy controls	MS progression	3.6 lesions/year ± 4.2 vs. 1.1 lesions/year ± 0.9, respectively; *p* = 0.03	Cortical lesion accrual was greater in participants with SPMS than with RRMS.	[37]
**Radiological predictors**	**Cortical lesions**	2022	Prospective study	20 RRMS patients, 13 SPMS patients, along with 10 age-matched healthy controls	EDSS changes	β = 0.5, *p* = 0.003	Total cortical lesion volume independently predicted baseline EDSS and EDSS changes at follow-up	[37]
**Radiological predictors**	**Cortical lesions**	2017	10-year observational, cross-sectional study	40 OCB-negative and 50 OCB-positive MS	Presence of cytokines	CXCL13 (r = 0.922; *p* < 0.001), CXCL12 (r = 0.678; *p* = 0.022), OPN (r = 0.692; *p* = 0.018), IL6 (r = 0.628; *p* = 0.039), TWEAK (r = 0.629; *p* = 0.038)	CL load significantly correlated with levels of several molecules linked to the B cell immune response.	[38]
**Fluid biomarkers**	**CSF biomarkers**	2020	Longitudinal 4-yearprospective study	99 RRMS patients (treatment-naive)	Risk of EDA (measures of disease activity: (1) evidence of relapses; (2) confirmed disability progression as assessed by an increase of the EDSS score by at least 1 point sustained over 6 months; and (3) evidence of new or newly enlarging WMT2 lesions).	HR = 1.78; *p* = 0.0001	CXCL13, LIGHT and APRIL were the CSF molecules more strongly associated with the risk of EDA.	[39]
**Fluid biomarkers**	**CSF biomarkers**	2020	Longitudinal 4-year prospective study	99 RRMS patients (treatment-naive)	Risk of cortical thinning.	β = 4.7 × 10^−4^; *p* < 0.001	Higher CSF levels of CXCL13 were associated with more severe cortical thinning.	[39]
**Fluid biomarkers**	**CSF biomarkers**	2024	Prospective study	118 de novo diagnosed RRMS patients and 112 controls	Correlation with number of T2 and Gd(+) lesions on head MRI in patients with newly diagnosed RRMS.	R Spearman: 0.2434; t (N-2) 2.2165; *p* = 0.0296	TNF-α levels positively correlated with post-contrast-enhancing brain lesions.	[40]
**Fluid biomarkers**	**CSF biomarkers**	2024	Prospective study	118 de novo diagnosed RRMS patients and 112 controls	Correlation with number of T2 and Gd (+) lesions on C-spine MRI in patients with newly diagnosed RRMS.	R Spearman: −0.2730; t (N-2) −2.5058; *p* = 0.0143	IL-15 levels in CSF correlated negatively with both the number of T2 lesions in C spine MRI and the number of Gd(+) lesions in C spine MRI	[40]
**Fluid biomarkers**	**CSF biomarkers**	2021	Meta-analysis	Six longitudinal studies, 1221 CIS/early RRMS patients	Risk of a second clinical relapse.	HR = 3.62, 95% CI 1.75–7.48, I2 = 88%, *p* = 0.0005	The pooled analysis confirmed that the presence of intrathecal IgM synthesis is a risk factor for a second clinical relapse.	[41]
**Demographic and clinical prognostic factors**	**Disease duration**	2023	Retrospective multicenter cohort study (RISM)	16,130 MS patients	PIRA	HR, 1.04; 95% CI, 1.04–1.05; *p* < 0.001	A longer disease duration was associated with a higher risk of PIRA events.	[26]
**Demographic and clinical prognostic factors**	**Disease duration**	2019	Retrospective cohort	2083 RRMS	Walking speed (T25FW speed)	HR −0.05 95% CI (−0.08, −0.02) < 0.001	Walking speed is slower in patients with a longer disease duration (per 5 years).	[18]
**Demographic and clinical prognostic factors**	**Disease duration**	2013	Retrospective study (London Multiple Sclerosis Clinic database)	730 MS patients	DSS 6	OR, 0.76 [95% CI, 0.69–0.84] and 0.44 [95% CI, 0.37–0.52] for 5- and 15-year latency, respectively)	Longer latency to progression was associated with lower probability of attaining DSS 6.	[42]
**Treatment**	**DMT exposure**	2023	Retrospective multicenter cohort study (RISM)	16,130 MS patients	PIRA	HR, 0.69; 95% CI, 0.64–0.74; *p* < 0.001	A shorter DMT exposure as associated with a higher risk of PIRA events.	[26]
**Treatment**	**DMT exposure**	2021	Retrospective 2-center cohort study	687 RRMS patients	RAW	sHR = 1.11 (1.02–1.21), *p* = 0.015	RAW was predicted by the temporary or permanent discontinuation of the initial DMT	[15]
**Treatment**	**DMT exposure**	2020	Retrospective multicenter cohort study (RISM)	646 POMS, 8473 AOMS and 382 LOMS patients at the first demyelinating event	Risk of 12-month confirmed disability worsening	aHR in non-exposed versus exposed: 6.3 (4.9–8.0) for adult-onset, *p* < 0.0001; LOMS 1.9 (0.9–4.1), *p* = 0.07.	DMT exposure reduced the risk of 12-month CDW, with a progressive risk reduction in different quartiles of exposure in paediatric-onset and adult-onset patients.	[43]
**Treatment**	**DMT exposure**	2020	Retrospective multicenter cohort study (RISM)	646 POMS, 8473 AOMS and 382 LOMS patients at the first demyelinating event	Risk of sustained EDSS 4.0	aHR in non-exposed versus exposed: 6.3 (4.9–8.0) for adult-onset, *p* < 0.0001; LOMS 1.9 (0.9–4.1), *p* = 0.07.	DMT exposure reduced the risk of sustained EDSS score of 4.0	[43]
**Treatment**	**DMT exposure**	2016	Retrospective multi-center cohort study	2466 MS patients	EDSS at 10 years	Coeff = −0.86, *p* = 1.3 × 10^−9^.	Cumulative treatment exposure was independently associated with lower EDSS at 10 years.	[44]
**Treatment**	**DMT exposure**	2020	Retrospective multi-center cohort study (MSBase registry)	1085 patients with ≥15-year follow-up	Risk of relapses	HR 0.59, 95% CI 0.50–0.70, *p* = 10^−9^	Treated patients were less likely to experience relapses (0.59, 0.50–0.70, *p* = 10^−9^) and worsening of disability	[45]
**Treatment**	**DMT exposure**	2020	Retrospective multi-center cohort study (MSBase registry)	1085 patients with ≥15-year follow-up	Risk of EDSS worsening	HR 0.81, 95% CI 0.67–0.99, *p* = 0.043	Treated patients were less likely to experience relapses (0.59, 0.50–0.70, *p* = 10^−9^) and worsening of disability	[45]
**Treatment**	**DMT exposure**	2020	Retrospective multicenter cohort study (RISM)	19,318 RRMS; 2343 SPMS identified with the DDA definition and 3868 identified with the neurologist	SPMS conversion (SPMS definition according to ND and DDA)	DDA group: HR (95% CI) 0.43 (0.36–0.50) *p* < 0.0001	A longer exposure to DMT is associated with lower risk of SPMS	[22]
**Other biomarkers**	**Evoked potentials**	2011	Retrospective monocentric study	80 MS patients	Risk of EDSS 4.0 and 6.0	log-rank test: *p* < 0.001	Increased risk of disability in patients with EP score higher than the median value. EP score of 8 or 9 showed the highest sensitivity and specificity in predicting EDSS 4.0 and 6.0	[46]
**Other biomarkers**	**Evoked potentials**	2023	Prospective monocentric study	181 MS patients	Risk of MSSS worsening	OR 0.04; IC 95% 0.01–0.06; *p*-Value 0.002	P100 latency resulted in a predictor for disability over time (MSSS).	[47]
**Other biomarkers**	**Evoked potentials**	2016	Retrospective monocentric study	100 MS patients	EDSS worsening from baseline data	OR = 1.2; 95 % CI 1.1–1.3; *p* = 0.0012	Baseline global EP score was a highly significant predictor of EDSS progression 6 years later.	[48]
**Fluid biomarkers**	**GFAP**	2022	Prospective cohort study (Comprehensive Longitudinal Investigation of MS at the Brigham and Women’s Hospital -climbstudy.org)	257 MS patients	6-months confirmed disability progression (6mCDP). EDSS progression was defined as an increase in the EDSS score since the previous visit of ≥1.0 point from an EDSS score of 1.0–5.0 or ≥0.5 point from an EDSS score of ≥5.5.6mCDP was defined as EDSS progression that was sustained for at least 180 days.	HR = 1.71; 95% CI = 1.19–2.45; *p* = 0.004	Higher sGFAP levels were associated with higher risk of 6mCDP. The association was stronger in patients with low sNfL (aHR = 2.44; 95% CI 1.32–4.52; *p* = 0.005) and patients who were nonactive in the 2 years prior or after the sample.	[49]
**Fluid biomarkers**	**GFAP**	2017	Retrospective monocentric study	GFAP levels in the CSF from 18 patients with RRMS, 8 patients with CIS and 35 healthy controls	Infratentorial chronic inflammatory lesion load	r = 0.55, *p* = 0.004	GFAP concentrations significantly correlated with infratentorial chronic, post-inflammatory lesion load	[50]
**Fluid biomarkers**	**GFAP**	2017	Retrospective monocentric study	GFAP levels in the CSF from 18 patients with RRMS, 8 patients with CIS and 35 healthy controls	Infratentorial chronic inflammatory lesion load	r = 0.71, *p* = 0.0002	GFAP concentrations significantly correlated with the intensity of gadolinium-enhancement as a parameter for the acute activity of inflammatory processes.	[50]
**Fluid biomarkers**	**GFAP**	2024	Retrospective study (but with prospective data collection)	133 RRMS patients	SPMS conversion	c β: 0.34 [−0.78;1.46]; *p* = 0.555	GFAP was not associated with conversion to SPMS.	[51]
**Fluid biomarkers**	**GFAP**	2024	Retrospective study (but with prospective data collection)	133 RRMS patients	EDSS score worsening	c β: 0.34 [−0.78;1.46]; *p* = 0.556	GFAP was not associated with disability progression.	[51]
**Radiological predictors**	**Gray matter pathology**	2014	Prospective cohort study	73 MS patients	EDSS worsening	OR = 0.79, *p* = 0.01; C-index = 0.69	Baseline GMF is predictor of worsening of disability in the long term.	[52]
**Radiological predictors**	**Gray matter pathology**	2021	Prospective study	332 MS patients, 96 healthy controls	Cognitive decline (test-defined assessment)	Nagelkerke R2 = 0.22, *p* < 0.001	A prediction model that included only whole-brain MRI measures showed cortical grey matter volume as the only significant MRI predictor of cognitive decline.	[53]
**Radiological predictors**	**Gray matter pathology**	2021	Prospective study	63 MS patients	EDSS	−0.26 (−0.444 to −0.074)	Across all subjects, cortical lesions, grey matter volume and cervical cord volume explained 60% of the variance of the Expanded Disability Status Scale.	[36]
**Radiological predictors**	**Gray matter pathology**	2022	Retrospective multi-center cohort study	373 MS patients	Difference in mean EDSS score over the years of follow-up	A deep learning architecture based on convolutional neural networks was implemented to predict: (1) clinical worsening (EDSS-based model), (2) cognitive deterioration (SDMT-based model), or (3) both (EDSS + SDMT-based model).	The convolutional neural network model showed high predictive accuracy for clinical (83.3%) and cognitive (67.7%) worsening, although the highest accuracy was reached when training the algorithm using both EDSS and SDMT information (85.7%).	[54]
**Radiological predictors**	**Gray matter pathology**	2014	Prospective study	81 MS patients	Disease progression	Patients with disability Progression showed significantly increased loss of whole brain (−3.8% vs. −2.0%, *p* < 0.001), cortical (−3.4% vs. −1.8%, *p* = 0.009) compared to patients with no progression.	GM atrophy showed association with disease progression	[55]
**Demographic and clinical prognostic factors**	**Onset type**	2022	Retrospective study	21 RRMS patients, 13 SPMS patients, along with 10 age-matched healthy controls	SPMS conversion	*p* > 0.05	Affected bowel and bladder functions during the first relapse were ineffective in predicting the transition to the SPMS course.	[56]
**Demographic and clinical prognostic factors**	**Onset type**	2015	Observational study based on a prospective, open cohort	1016 CIS	Risk of reaching EDSS score of at least 3.0 in 2 evaluations (defined “disability accumulation”)	HR 0.5; 95% CI 0.3–0.8	Patients presenting CIS with optic neuritis appeared to display a lower risk of reaching an EDSS score of 3.0.	[33]
**Demographic and clinical prognostic factors**	**Onset type**	2016	Prospective observational study	793 MS patients	Reaching moderate disability: EDSS 3.0–5.5	HR 0.42; CI 0.23–0.77; *p* = 0.005	Complete remission of neurological symptoms at onset reduced the risk of moderate disability.	[35]
**Demographic and clinical prognostic factors**	**Onset type**	2020	Retrospective multicenter cohort study (RISM)	19,318 RRMS; 2343 SPMS identified with dhe DDA definition and 3868 identified with the neurologist definition (ND)	SPMS conversion (SPMS definition according to ND and DDA)	DDA group: HR (95% CI) 1.26 (1.12–1.40), *p* < 0.0001; ND group: 1.13 (1.03–1.23), *p* = 0.011	Multifocal onset is associeted with higher risk of SPMS	[22]
**Demographic and clinical prognostic factors**	**Onset type**	2020	Retrospective monocentric study	157 PPMS patients	Time to EDSS 6.0	HR (95% CI): 2.13 (1.24–3.63); *p*= 0.006	The presence of spinal motor symptoms at onset were associated with a shorter time to EDSS6	[25]
**Demographic and clinical prognostic factors**	**Onset type**	2013	Retrospective monocentric study	197 MS patients	Risk of EDSS 6.0	8.1 and 13.1 fold increased risk to EDSS 6, respectively (*p* = 0.04 and *p* = 0.01).	Motor and brainstem symptoms at onset were also associated with higher risk of EDSS 6.0	[57]
**Demographic and clinical prognostic factors**	**Onset type**	2020	Prospective study	415 MS patients	Risk of EDSS 6.0	HR 2.107, 95% CI 1.168–3.800, *p* = 0.013	An incomplete recovery from first attack was significantly associated with a higher HR to reach an EDSS of 6.0.	[27]
**Demographic and clinical prognostic factors**	**Onset type**	2015	Population-based cohort (retrospective - prospective)	Population-based cohort (105 patients with relapsing-remitting MS, 86 with bout-onset progressive MS) and a clinic-based cohort (415 patients with bout-onset progressive MS)	Recovery from first relapse	*p* = 0.001	A brainstem, cerebellar, or spinal cord syndrome was associated with a poor recovery from the initial relapse.	[58]
**Other biomarkers**	**Optic nerve diameter**	2021	Prospective study	63 MS patients	Disease progression	*p* = 0.041 for the right eye and *p* = 0.037 for the left eye	Smaller diameters of optic nerve are associated with poor clinical progression and greater disability (measured by EDSS).	[59]
**Other biomarkers**	**Optic nerve diameter**	2021	Prospective study	63 MS patients	Sustained increase (>3 months) of over 0.5 points on the EDSS.	*p* = 0.07 for the right eye and *p* = 0.043 for the left eye	Smaller diameters of optic nerve are associated with poor clinical progression and greater disability (measured by EDSS).	[59]
**Other biomarkers**	**Optic nerve diameter**	2019	Prospective study	49 RRMS patients, 50 matched healthy controls	Sustained EDSS > 2	*p* = 0.044, OR = 0.000, 95% CI = 0.000–0.589	Optic nerve diameter was an independent predictor of EDSS > 2	[60]
**Demographic and clinical prognostic factors**	**PIRA**	2023	Retrospective analysis of data from patients prospectively included (patients with a first demyelinating attack)	1128 patients	Adjusted yearly EDSS increase rates	0.31; 95% CI, 0.26–0.35 vs. 0.13; 95% CI, 0.10–0.16; *p* < 0.001	Early PIRA had steeper EDSS yearly increase rates than late PIRA.	[23]
**Demographic and clinical prognostic factors**	**PIRA**	2023	Retrospective analysis of data from patients prospectively included (patients with a first demyelinating attack)	1128 patients	Risk of reaching EDSS 6.0	HR, 26.21; 95% CI, 2.26–303.95; *p* = 0.009	Early PIRA had a 26-fold greater risk of reaching EDSS 6.0 from the first attack (HR, 26.21; 95% CI, 2.26–303.95; *p* = 0.009).	[23]
**Demographic and clinical prognostic factors**	**PIRA**	2023	Retrospective analysis of data from patients prospectively included (patients with a first demyelinating attack)	1128 patients	Risk of reaching EDSS 6.0	HR, 7.93; 95% CI, 2.25–27.96; *p* = 0.001	Patients with PIRA had an 8-fold greater risk of reaching EDSS 6.0.	[23]
**Radiological predictors**	**Presence of new Gd + SC lesions**	2018	Single-centre retrospective study	201 RRMS patients	Relapse occurrence (clinical relapses within 3 months)	B 1.113, Exp (B), 95% CI for EXP(B) 3.042, 1.158–7.995; *p* = 0.024	A significant association between new Gd + SC lesions and clinical relapses within 3 months was found.	[61]
**Radiological predictors**	**Presence of new Gd + SC lesions**	2018	Single-centre retrospective study	201 RRMS patients	DMT changes within 3 months	B 1.482, Exp (B), 95% CI for EXP(B) 4.402, 1.642–11.799; *p* = 0.003	Even without clinical symptoms, worsening SC findings significantly predicted treatment changes.	[61]
**Fluid biomarkers**	**Presence of OCBs**	2013	Meta analysis: 71 studies	12,253 MS patients,	EDSS worsening, EDSS disability milestones	1.96 (95% CI 1.31 to 2.94; *p* = 0.001) with no between-study heterogeneity (I2 = 0%; X2 = 2.95, df = 3, *p* = 0.40)	OCB-positive MS patients had an OR of 1.96 of reaching disability outcomes.	[62]
**Fluid biomarkers**	**Presence of OCBs**	2021	Retrospective registry-based study	7322 patients, 6494 OCB+	Risk of reaching sustained EDSS score milestones 3.0, 4.0 and 6.0	EDSS 3.0 (HR = 1.29, 95% CI 1.12 to 1.48, *p* < 0.001) and 4.0 (HR = 1.38, 95% CI 1.17 to 1.63, *p* < 0.001).	CSF-OCB presence is associated with higher risk of reaching EDSS milestones 3.0 anf 4.0.	[63]
**Fluid biomarkers**	**Presence of OCBs**	2021	Retrospective registry-based study	7322 patients, 6494 OCB+	SPMS conversion.	HR: 1.20, 95% CI 1.02 to 1.41, *p* = 0.03, n = 5721	OCB positivity IS associated with increased risk of conversion to SPMS.	[63]
**Fluid biomarkers**	**Presence of OCBs**	2015	Observational study based on a prospective, open cohort	1017 CIS	Risk of reaching EDSS score of at least 3.0 in 2 evaluations (defined “disability accumulation”)	adjusted HR scores of 2.0 (95% CI 1.2–3.6)	The presence of OCBs was associated with a higher risk of the accumulation of disability	[33]
**Fluid biomarkers**	**Presence of OCBs**	2021	Meta-analysis	Six longitudinal studies, 1221 CIS/early RRMS patients	Risk of a second clinical relapse.	HR = 2.18, 95% CI 1.24–3.82, I2 = 73%, *p* = 0.007	The pooled analysis confirmed that the presence of OCBs (IgG) is a risk factor for a second clinical relapse.	[41]
**Fluid biomarkers**	**Presence of OCBs**	2021	Retrospective monocentric study	358 patients, 287 OCB positive	MSSS	OCB + vs. OCB - (2.10 vs. 0.94, *p* value = 0.023)	Median MSSS was significantly higher in the OCB positive group (2.10 vs. 0.94, *p* value = 0.023) and remained significant when controlling for age at EDSS.	[64]
**Fluid biomarkers**	**Presence of OCBs**	2017	10-year observational, cross-sectional study	40 OCB-negative and 50 OCB-positive MS	Presence of cortical lesions	mean ± standard deviation: OCB + 6.1 ± 6.1 (0–24), ocb − 2.2 ± 2.8 (0–11), *p* < 0.0001	Increased number of CLs was found in OCB+ compared to OCB− patients.	[38]
**Demographic and clinical prognostic factors**	**Relapses**	2015	Population-based cohort (retrospective-prospective)	Population-based cohort (105 patients with relapsing-remitting MS, 86 with bout-onset progressive MS) and a clinic-based cohort (415 patients with bout-onset progressive MS)	SPMS conversion	Half of the good recoverers developed progressive MS by 30.2 years after MS onset, whereas half of the poor recoverers developed progressive MS by 8.3 years after MS onset (*p* = 0.001).	Patients with MS with poor recovery from early relapses will develop progressive disease course earlier than those with good recovery.	[58]
**Demographic and clinical prognostic factors**	**Relapses**	2020	Prospective study	415 MS patients	Risk of EDSS 6.0	HR 2.217, 95% CI 1.148–4.281, *p* = 0.018	≥2 relapses during the first 2 years after onset were significantly associated with a higher HR to reach an EDSS of 6.0.	[27]
**Demographic and clinical prognostic factors**	**Relapses**	2015	Population-based cohort (retrospective-prospective)	Population-based cohort (105 patients with relapsing-remitting MS, 86 with bout-onset progressive MS) and a clinic-based cohort (415 patients with bout-onset progressive MS)	Recovery from first relapse	*p* = 0.001	A fulminant relapse was associated with a poor recovery from the initial relapse.	[58]
**Demographic and clinical prognostic factors**	**Relapses**	2010	Retrospective study (London Multiple Sclerosis Clinic database)	806 RTMS patients	DSS 6, 8, 10	various OR	Frequent relapses in the first 2 years and shorter first inter-attack intervals predicted shorter times to reach hard disability endpoints.	[65]
**Demographic and clinical prognostic factors**	**Relapses**	2013	Retrospective monocentric study	197 MS patients	Risk of EDSS 8.0	1.28 (5 years) and 1.19 (10 years), respectively *p* = 0.032 and *p* = 0.015	The number of relapses in five and ten years of disease onset was associated with a slightly increased risk to EDSS 8	[57]
**Demographic and clinical prognostic factors**	**Relapses**	2024	Retrospective cohort	115 RMS	Change in EDSS from first assessment at relapse to EDSS after the last relapse treatment for all relapse events: EDSS improvement (after relapse treatment with steroid or PLEX)	Regression coefficient (95% CI): −0.32 (−0.44, −0.19) *p* < 0.001	Female sex, younger age, and a higher EDSS during relapse as factors associated with a higher chance of EDSS improvement after relapse treatment.	[17]
**Demographic and clinical prognostic factors**	**Relapses**	2020	Retrospective multicenter cohort study (RISM)	19,318 RRMS; 2343 SPMS identified with the DDA definition and 3868 identified with the neurologist definition (ND)	SPMS conversion (SPMS definition according to ND and DDA)	DDA group: HR (95% CI) 2.90 (2.54–3.30), *p* < 0.0001; ND group: 1.78 (1.64–1.94), *p* < 0.0001	A higher number of relapses during RRMS phase is associated with higher risk of SPMS	[22]
**Demographic and clinical prognostic factors**	**Relapses**	2020	Retrospective multicenter cohort study (RISM)	646 POMS, 8473 AOMS and 382 LOMS patients at the first demyelinating event	Risk of 12-month confirmed disability worsening	aHR: AOMS 1.37 (1.36–1.39); LOMS 1.40 (1.31–1.49)	Relapses were a risk factor for 12-month confirmed disability worsening in all three cohorts	[43]
**Fluid biomarkers**	**Serum NfL level**	2020	Prospective cohort study	258 MS patients	Conversion to clinically diagnosed progressive MS	AUC of 0.744 (95% CI 0.61–0.88, *p* = 0.054).	MS patients with low serum NfL values (<7.62 pg/mL) at the baseline were 7.1 times less likely to develop progressive MS.	[66]
**Fluid biomarkers**	**Serum NfL level**	2020	Prospective cohort study	258 MS patients	Annual rate of EDSS progression	0.17 units/year, Kruskal–Wallis *p* = 0.020, df 2	Patients with the highest NfL levels (>13.2 pg/mL) progressed most rapidly with an EDSS annual rate of 0.16 (*p* = 0.004), remaining significant after adjustment for sex, age, and disease-modifying treatment (*p* = 0.022)	[66]
**Fluid biomarkers**	**Serum NfL level**	2022	Case-control	5390 control, 1313 MS patients	Risk of relapse	OR 1.41, 95% CI 1.30–1.54; *p* < 0.0001	Patients with higher sNfL Z scores showed a greater probability of relapses in the following year, based on a model with Z score as a continuous predictor.	[67]
**Fluid biomarkers**	**Serum NfL level**	2022	Case-control	5390 control, 1313 MS patients	EDSS worsening	OR 1.11, 1.03–1.21; *p* = 0.0093	People with higher sNfL Z scores showed a greater probability of EDSS worsening in the following year, based on a model with Z score as a continuous predictor.	[67]
**Fluid biomarkers**	**Serum NfL level**	2022	Case-control	5390 control, 1313 MS patients	EDA 3	OR 1.43, 1.31–1.57; *p* < 0.0001	People with higher sNfL Z scores showed a greater probability of EDA-3 in the following year, based on a model with Z score as a continuous predictor	[67]
**Fluid biomarkers**	**Serum NfL level**	2022	Case-control	5390 control, 1313 MS patients	Clinical or MRI disease activity	All people with multiple sclerosis: OR 3.15, 95% CI 2.35–4.23; *p* < 0.0001); people considered stable with no evidence of disease activity (2.66, 1.08–6.55; *p* = 0.034)	A sNfL Z score above 1.5 was associated with an increased risk of future clinical or MRI disease activity in all people with multiple sclerosis and in people considered stable with no evidence of disease activity.	[67]
**Fluid biomarkers**	**Serum NfL level**	2019	Prospective study	235 MS patients in a 2-year RCT of intramuscular interferon β-1a, and in serum (n = 164) from the extension study.	Risk of EDSS ≥ 6.0 at Year 8	OR = 3.4; 95% CI = 1.2–9.9, *p* < 0.05	Year 2 CSF levels were predictive of reaching EDSS ≥ 6.0 at Year 8	[68]
**Fluid biomarkers**	**Serum NfL level**	2019	Prospective study	235 MS patients in a 2-year RCT of intramuscular interferon β-1a, and in serum (n = 164) from the extension study.	Risk of EDSS ≥ 6.0 at Year 8	OR = 11.0, 95% CI = 2.0–114.6; *p* < 0.01	Year 3 serum levels were predictive of reaching EDSS ≥ 6.0 at Year 9	[68]
**Fluid biomarkers**	**Serum NfL level**	2019	Prospective study	235 MS patients in a 2-year RCT of intramuscular interferon β-1a, and in serum (n = 164) from the extension study.	Risk of EDSS ≥ 6.0 at Year 15	OR (upper vs. lower tertile) = 4.9; 95% CI = 1.4–20.4; *p* < 0.05	Year 4 serum levels were predictive of reaching EDSS ≥ 6.0 at Year 15	[68]
**Fluid biomarkers**	**Serum NfL level**	2021	Prospective study	369 blood samples from 155 early relapsing-remitting MS patients on interferon beta-1a.	Odds of EDA-3	upper vs. lower 86.5% vs. 57.9%; OR = 4.25, 95% CI: [2.02, 8.95]; *p* = 0.0001	In patients with disease activity (EDA-3), those with higher sNFL had higher odds of EDA-3 in the following year than those with low sNFL.	[69]
**Fluid biomarkers**	**Serum NfL level**	2021	Prospective study	369 blood samples from 155 early relapsing-remitting MS patients on interferon beta.	BVL (brain volume loss)	β = −0.36%; 95% CI = [−0.60, −0.13]; *p* = 0.002	In patients with disease activity (EDA-3), those with higher sNFL had greater whole brain volume loss during the following year.	[69]
**Fluid biomarkers**	**Serum NfL level**	2020	Prospective cohort study	258 MS patients	EDSS score of ≥ 4	2nd tertile (>7.62 pg/mL): HR = 5.5 (95% CI 1.4–21.0), *p* = 0.012; 3rd tertile (>13.2 pg/mL): HR = 5.2 (95% CI 1.5–18.6), *p* = 0.010. AUC of 0.734 (95% CI 0.63–0.84, *p* = 0.001).	MS patients with higher serum NfL values (>7.62 pg/mL) at the baseline had a significantly higher risk of developing an EDSS ≥ 4, showing that they were on average > 5-times at higher risk of developing EDSS ≥ 4 over the follow-up.	[66]
**Fluid biomarkers**	**Serum NfL level**	2024	Retrospective study (but with prospective data collection)	133 RRMS patients	SPMS conversion	c β [95% CI]: 9.92 [0.62;19.21]; *p* = 0.037	sNfL was associated with conversion to SPMS,	[51]
**Fluid biomarkers**	**Serum NfL level**	2024	Retrospective study (but with prospective data collection)	133 RRMS patients	EDSS score worsening	c β: 0.34 [−0.78;1.46]; *p* = 0.554	sNfL was not associated with disability progression.	[51]
**Demographic and clinical prognostic factors**	**Sex**	2013	Retrospective multi-center cohort study (MSBase registry)	11,570 relapse-onset patients and 881 patients with PPMS	Relapse incidence	Relapse frequency was 17.7% higher in females compared with males.	Within the initial 5 years, the female-to-male ratio increased from 2.3:1 to 3.3:1 in patients with 0 versus ≥4 relapses per year, respectively.	[14]
**Demographic and clinical prognostic factors**	**Sex**	2015	Observational study based on a prospective, open cohort	1015 CIS	Risk of reaching EDSS score of at least 3.0 in 2 evaluations (defined “disability accumulation”)	HR 0.7; 95% CI 0.5–1.1	Female sex appeared to display a lower risk of reaching an EDSS score of 3.0.	[33]
**Demographic and clinical prognostic factors**	**Sex**	2024	Retrospective cohort	113 RMS	Change in EDSS from first assessment at relapse to EDSS after the last relapse treatment for all relapse events: EDSS improvement (after relapse treatment with steroid or PLEX)	Regression coefficient (95% CI): −0.60 (−0.98, −0.21) *p* < 0.01	Female sex, younger age, and a higher EDSS during relapse as factors associated with a higher chance of EDSS improvement after relapse treatment.	[17]
**Demographic and clinical prognostic factors**	**Sex**	2019	Retrospective cohort	2083 RRMS	Walking speed (T25FW speed)	HR 0.28, 95% CI 0.20–0.36, <0.001	Walking speed is slower in females.	[18]
**Demographic and clinical prognostic factors**	**Sex**	2013	Retrospective monocentric study	197 MS patients	Risk of EDSS 6.0, 7.0	4.63-fold increased risk to EDSS 6 (*p* < 0.001); 4.69-fold increased risk to EDSS 7 (*p* = 0.006).	Male sex was associated with a higher risk to EDSS 6 and 7.	[57]
**Demographic and clinical prognostic factors**	**Sex**	2024	Prospective study	149 MS patients	time-to-relapse	HR = 0.91; 95 %CI = 0.53–1.58	No sex differences in time-to-relapse emerged.	[70]
**Demographic and clinical prognostic factors**	**Sex**	2024	Prospective study	149 MS patients	EDSS worsening	OR = 0.75; 95% CI = 0.21–2.35	Males had no increased risk of EDSS worsening compared to females.	[70]
**Demographic and clinical prognostic factors**	**Sex**	2020	Retrospective multicenter cohort study (RISM)	646 POMS, 8473 AOMS and 382 LOMS patients at the first demyelinating event	Risk of 12-month confirmed disability worsening	aHR: LOMS, female sex 0.74 (0.53–1.04)	Female sex exerted a protective role in the late-onset cohort for the risk of confirmed 12-months CDW.	[43]
**Radiological predictors**	**Spinal cord atrophy**	2014	Prospective study	159 MS patients	Risk of EDSS score of at least 6.0 (requirement of a walking aid)	OR = 0.57 per 1 SD higher cord area; 95% CI 0.37, 0.86; *p* = 0.01	Long-term physical disability was independently linked with atrophy of the spinal cord	[71]
**Radiological predictors**	**Spinal cord atrophy**	2021	Prospective study	63 MS patients	EDSS	−0.27 (−0.421 to −0.109)	Across all subjects, cortical lesions, grey matter volume and cervical cord volume explained 60% of the variance of the Expanded Disability Status Scale.	[36]
**Radiological predictors**	**Spinal cord lesions**	2024	Monocentric retrospective study	205 RRMS	CDA occurrence. CDA was defined as an EDSS increase of 1.5 for baseline EDSS scores of 0, an increase of 1 for baseline EDSS scores between 1.0 and 5.0, and an increase of 0.5 for baseline EDSS scores of 5.5 or higher; the increase had to be confirmed on clinical follow-up over at least 6 months; PIRA and RAW	Spearman’s rank correlation coefficient (rs): SCLN and SCLV were closely correlated (rs = 0.91, *p* < 0.001) and were both significantly associated with CDA on follow-up (*p* < 0.001). Subgroup analyses confirmed this association for patients with PIRA on CDA (34 events, *p* < 0.001)	The number of SC lesions on MRI is associated with future accumulation of disability largely independent of relapses.	[72]
**Radiological predictors**	**Spinal cord lesions**	2019	Prospective study	178 MS patients	EDSS correlation	β = 1.53, *p* < 0.01	Spinal cord lesions showed a consistent association with Expanded Disability Status Scale at 15 years.	[32]
**Radiological predictors**	**Spinal cord lesions**	2019	Prospective study	178 MS patients	SPMS conversion at 15 years follow-up	OR 4.71, 1.72, 12.92; *p* = 0.003	Spinal cord lesions were independently associated with secondary progressive multiple sclerosis at 15 years.	[32]
**Radiological predictors**	**Spinal cord lesions**	2021	Retrospective 2-center cohort study	687 RRMS patients	CDA occurrence	Sub distribution HR = 4.08 (1.29–12.87), *p* = 0.016	The risk of PIRA was associated with the presence of spinal cord lesions at baseline MRI scan.	[15]
**Radiological predictors**	**Spinal cord lesions**	2024	Monocentric retrospective study	204 RRMS	CDA occurrence	OR 5.8, 95% CI 2.1 to 19.10	The volume of SC lesions on MRI is associated with future accumulation of disability largely independent of relapses.	[72]
**Radiological predictors**	**Spinal cord lesions**	2010	Retrospective cohort study	25 RRMS patients	EDSS 4.0	HR 7.2, 95% confidence interval 1.4–36.4	The diffuse abnormality in cervical spinal cord at the beginning of the disease is persistent and predicts a worse prognosis in RRMS patients.	[73]
**Radiological predictors**	**Spinal cord lesions**	2024	Monocentric retrospective study	204 RRMS	CDA occurrence	OR 5.8, 95% CI 2.1 to 19.8	Patients without any SC lesions experienced significantly less CDA	[72]
**Treatment**	**Time to DMT initiation**	2023	Retrospective multicenter cohort study (RISM)	16,130 MS patients	Risk of PIRA	HR, 1.16; 95% CI, 1.00–1.34; *p* = 0.04	Delayed DMT initiation was associated with higher risk of PIRA events.	[26]
**Treatment**	**Time to DMT initiation**	2023	Retrospective multicenter cohort study (RISM)	16,130 MS patients	Risk of RAW	HR, 1.75; 95% CI, 1.28–2.39; *p* = 0.001	Delayed DMT initiation was associated with higher risk of RAW events.	[26]
**Treatment**	**Time to DMT initiation**	2021	Retrospective multi-center cohort study (BMSD)	11,871 RRMS patients	3-month CDW	Retrospective 2-center cohort study+G91	The time interval between disease onset and the first DMT start is a strong predictor of disability accumulation, independent of relapse activity, over the long-term.	[74]
**Treatment**	**Time to DMT initiation**	2021	Retrospective multi-center cohort study (BMSD)	11,871 RRMS patients	12-month CDW	HR 95% CI 1.21 (1.09–1.35) *p* = 0.0004	The time interval between disease onset and the first DMT start is a strong predictor of disability accumulation, independent of relapse activity, over the long-term.	[74]
**Treatment**	**Time to DMT initiation**	2020	Retrospective multi-center cohort study (MSBase registry; Swedish MS registry)	308 in the MSBase registry and 236 in the Swedish MS registry	Difference in mean EDSS score over the years of follow-up	Mean EDSS 2.2 (SD 1.6) in the early group compared with 2.9 (SD 1.8) in the late group (*p* < 0.0001). All follow-up years (mean EDSS score 2.3 [SD 1.8] vs. 3.5 [SD 2.1]; *p* < 0.0001), with a difference between groups of −0.98 (95% CI −1.51 to −0.45; *p* < 0.0001)	High-efficacy therapy commenced within 2 years of disease onset is associated with less disability after 6–10 years than when commenced later in the disease course.	[75]
**Treatment**	**Time to DMT initiation**	2018	Retrospective multi-center cohort study	3795 MS patients (Danish MS Register)	Risk of reaching EDSS 6.0	HR, 1.42; 95% confidence interval (CI), 1.18–1.70; *p* < 0.001	Patients who started treatment with DMT later reached an EDSS score of 6 more quickly compared with patients who started early	[76]
**Treatment**	**Time to DMT initiation**	2018	Retrospective multi-center cohort study	3795 MS patients (Danish MS Register)	Mortality	HR, 1.38; 95% CI, 0.96–1.99; *p* = 0.08	Mortality increased by 38% in later DMT starters.	[76]
**Treatment**	**Time to DMT initiation**	2023	Observational cohort study	2648 MS patients	MSIS physical score and psychological score worsening	Worsening of MSIS physical score worsening by 2.75 points (95% CI 1.29 to 4.20), and MSIS psychological score by 2.02 points (95% CI 0.03 to 3.78)	Earlier commencement of disease-modifying treatment was associated with better patient-reported physical symptoms	[77]
**Treatment**	**Treatment strategy**	2021	Retrospective multicenter cohort study (RISM)	2702 RRMS patients (PS matching: 363 pairs)	Mean annual delta-EDSS	Mean annual delta-EDSS values were all significantly (*p* < 0.02) higher in the ESC group compared with the EIT group. In particular, the mean delta-EDSS differences between the two groups tended to increase from 0.1 (0.01–0.19, *p* = 0.03) at 1 year to 0.30 (0.07–0.53, *p* = 0.009) at 5 years and to 0.67 (0.31–1.03, *p* = 0.0003) at 10 years.	EIT strategy is more effective than ESC strategy in controlling disability progression over time.	[78]
**Treatment**	**Treatment strategy**	2023	Retrospective multi-center cohort study (Swedish MS registry, Czech national MS registry)	6410 MS patients	Risk of reaching EDSS 4	HR 0.74, 95% CI 0.6–0.91, *p*-value 0.0327)	The risk of reaching EDSS 4.0 was reduced by 26% in patients starting with HE DMTs.	[79]
**Treatment**	**Treatment strategy**	2023	Retrospective multi-center cohort study (Swedish MS registry, Czech national MS registry)	6410 MS patients	Risk of relapses	HR 0.34, 95% CI 0.3–0.39, *p*-value < 0.001)	The risk of relapses was reduced by 66% (HR 0.34, 95% CI 0.3–0.39, *p*-value < 0.001)	[79]
**Treatment**	**Treatment strategy**	2023	Retrospective multi-center cohort study (Swedish MS registry, Czech national MS registry)	6410 MS patients	Probability of confirmed disability improvement (CDI)	HR 3.04, 95% CI 2.37–3.9, *p*-value < 0.001	The probability of CDI was three times higher.	[79]
**Treatment**	**Treatment strategy**	2019	Retrospective cohort study	592 MS patients	Mean (SD) 5-year change in EDSS score	EIT group vs. the ESC group (0.3 [1.5] vs. 1.2 [1.5]). β = −0.85; 95% CI, −1.38 to −0.32; *p* = 0.002)	Mean (SD) 5-year change in EDSS score was lower in the EIT group than the ESC group	[80]
**Treatment**	**Treatment strategy**	2019	Retrospective cohort study	593 MS patients	Median time to sustained accumulation of disability (SAD)	6.0 (3.17–9.16) years for EIT and 3.14 (2.77–4.00) years for ESC (*p* = 0.05).	Median time to SAD was longer for the EIT group	[80]
**Treatment**	**Treatment strategy**	2020	Retrospective cohort study (Danish MS Register)	388 patients in the study: 194 starting initial therapy with heDMT matched to 194 patients starting meDMT.	6-month confirmed EDSS score worsening	16.7% (95% CI 10.4–23.0%) and 30.1% (95% CI 23.1–37.1%) for heDMT and meDMT initiators, respectively (HR 0.53, 95% CI 0.33–0.83, *p* = 0.006).	A lower probability of 6-month confirmed EDSS score worsening was found in patients starting a heDMT as first therapy, compared to a matched sample starting meDMT.	[81]
**Treatment**	**Treatment strategy**	2020	Retrospective cohort study (Danish MS Register)	388 patients in the study: 194 starting initial therapy with heDMT matched to 194 patients starting meDMT.	Risk of first relapse after treatment start	HR 0.50, 95% CI 0.37–0.67	A lower probability of a first relapse was found in patients starting a heDMT as first therapy, compared to a matched sample starting meDMT.	[81]

## Data Availability

Not applicable.

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
