# Peer review of "Refining Prognostic Factors in Adult-Onset Multiple Sclerosis: A Narrative Review of Current Insights"

_ijms, 2025, doi:10.3390/ijms26167756_

Round 1

Reviewer 1 Report

Comments and Suggestions for Authors

I have read a high-quality and comprehensive manuscript. I would like to offer a few minor comments that may help to further improve the work:

- The paper may aid in selecting from an expanding range of therapies either at treatment initiation or when signs of suboptimal therapeutic response emerge.

- From the perspective of clinicians, much of the information compiled here reflects current real-world practices and informs day-to-day therapeutic decisions.

- The utility of oligoclonal bands is predominantly diagnostic and differential rather than prognostic, as they are present in over 90% of multiple sclerosis cases.

- It would be worthwhile to emphasize which prognostic factors are applicable in everyday clinical practice and which currently hold primarily academic or theoretical relevance.

Author Response

Response to Reviewer 1 Comments

The Reviewer's time spent reviewing this manuscript is greatly appreciated. We sincerely appreciate the Reviewer’s insightful comments, which reflect a clear understanding of the aims of our work: to integrate the most recent prognostic factors into a clinically oriented and up-to-date framework, with the ultimate goal of supporting therapeutic decisions and disease management in everyday practice.

Comment 1. The utility of oligoclonal bands is predominantly diagnostic and differential rather than prognostic, as they are present in over 90% of multiple sclerosis cases.

We appreciate and acknowledge the Reviewer's observation. Stressing, as the Reviewer suggest, the primary role of OCBs in diagnosis (Historically, CSF-specific oligoclonal band (OCB) testing has been the most generally accessible laboratory test recognized as the cornerstone of MS diagnosis [66]), we decided to include also the prognostic studies on OCBs because in the process of literature research and selection this topic emerged in several studies. Therefore, we expanded the section with a brief discussion on these.

Comment 2. It would be worthwhile to emphasize which prognostic factors are applicable in everyday clinical practice and which currently hold primarily academic or theoretical relevance.

We thank the reviewer for this wise suggestion, allowing us to better characterize the clinical application of our narrative review.  In line with other peer-reviewed comments, we have renamed the Section 8 to better reflect the focus of the discussion on prognostic tools: “The Strategic Role of Prognostic Algorithms in Clinical Decision-Making and Research.” This change aligns with the aim of the section, which was specifically intended to address clinical prognostic tools and synthesize those that can be used in everyday clinical practice by neurologists. We included and described another very recent, validated and clinical versatile tool, the Barcelona Risk Score (BRS). We stressed also a practical recommendation, in light of the discussion about prognosis throughout the text: In conclusion, our practical recommendation is to adopt integrative algorithms capable of assessing the patient comprehensively, by combining biomarkers that reflect the disease from multiple angles, ranging from clinical presentation to molecular indicators of progression, and neuroimaging findings.

Reviewer 2 Report

Comments and Suggestions for Authors

The review about prognostic factors for MS is a timely and useful narrative of current concepts about prognosis which is the main concern of MS patients; then, this review would be an interesting text, I suggest publication.  A minor point would be on the title, the line “a narrative review of current insights” is long and odd, I think that a simple term like “a review” would be sufficient, however, the editor could address this suggestion.

Also, supplemental material seems too long; however, only Table 1 (reduced) might be relevant and could be briefly explained in the text as was the case for Figure 1.

Author Response

Response to Reviewer 2 Comments

The Reviewer's time spent reviewing this manuscript is greatly appreciated, and we sincerely appreciate the Reviewer’s insightful comments.

Comment 1. A minor point would be on the title, the line “a narrative review of current insights” is long and odd, I think that a simple term like “a review” would be sufficient, however, the editor could address this suggestion.

We thank the Reviewer for the suggestion. However, we chose to include the term “narrative” in the title to accurately reflect the methodological approach of this review, and we believe that specifying its narrative nature is important for clarity and transparency.

Comment 2. Also, supplemental material seems too long; however, only Table 1 (reduced) might be relevant and could be briefly explained in the text as was the case for Figure 1.

We thank the reviewer for this wise suggestion. We have provided a more detailed description of the structure and variables presented in Table 1 in paragraph 2.2 of the Methods section. Although we agree that the table is quite long, we prefer to include it in full in order to highlight the process of analyzing and selecting the studies.

Reviewer 3 Report

Comments and Suggestions for Authors

The manuscript titled "Refining prognostic factors in adult-onset multiple sclerosis: a narrative review of current insights" addresses an important topic in the field of neurology and autoimmune diseases. Overall, the manuscript provides a comprehensive overview, though it needs some improvements before possible publication in the journal.

  1. The abstract is too brief and does not sufficiently convey the full scope or significance of the study.
  2. In introduction section the manuscript labels itself a “narrative review,” yet fails to justify why this approach was chosen over a systematic review.
  3. Key concepts like “PIRA,” “RAW,” and “smoldering MS” are introduced with some explanation in introduction section but lack precise definitions or clinical thresholds. These should be more clearly delineated for clarity and scientific rigor.
  4. Although the final sentences of introduction hint at the review’s purpose, the aim is still unclear. The authors should explicitly state the key questions being addressed and clarify the scope to sharpen the reader’s expectations.
  5. Time frame of collected studies should be mentioned in 2.2 section.
  6. Several terms in the manuscript are introduced in their abbreviated form without first being defined. For clarity and to ensure accessibility for a broad scientific readership, all abbreviations should be clearly defined at their first mention.
  7. Grammatical refinements are needed throughout the manuscript to enhance clarity and readability.

Author Response

Response to Reviewer 3 Comments

The Reviewer's time spent reviewing this manuscript is greatly appreciated. Please find the detailed responses below and the corresponding revisions highlighted in track changes in the re-submitted files.

Comment 1. The abstract is too brief and does not sufficiently convey the full scope or significance of the study.

We thank the Reviewer for the scrutiny, allowing us to improve several parts of our manuscript. We have now revised and entirely rewritten the abstract, to convey the full scope of the narrative review.

Multiple sclerosis (MS) is characterized by a continuum of diverse neuroinflammatory and neurodegenerative processes that contribute to disease progression from the earliest stages. This leads to a highly heterogeneous clinical course, requiring early and accurate prognostic assessment: the identification of reliable prognostic biomarkers is crucial to support therapeutic decision-making and guide personalized disease management. In this narrative review, we critically examined the current MS literature, investigating prognostic factors associated with disease progression and irreversible disability in adult-onset MS, with a focus on different clinical, radiological, and molecular biomarkers. Particular attention is direct toward the prognostic value of baseline clinical and neuroimaging factors, emerging biomarkers of smouldering disease, and progression independent of relapse activity (PIRA) events. Additionally, we discussed the role of integrated prognostic tools and risk scores, as well as their potential impact on clinical practice. We aim to provide a comprehensive and clinically oriented synthesis of available evidence in the MS biomarkers field, supporting multifaceted prognostication strategies to improve long-term outcomes in people with MS.

Comment 2. In introduction section the manuscript labels itself a “narrative review,” yet fails to justify why this approach was chosen over a systematic review.

We acknowledge the Reviewer's observation. We chose a narrative review format rather than a systematic review because our aim was to provide a comprehensive overview of the heterogeneous body of literature on prognostic factors in MS, integrating findings from different domains (clinical, molecular, MRI, etc.) and across patient subgroups. In the great heterogeneity of sources lies our methodological choice: considering different study designs, endpoints, and patient cohorts, a systematic review with meta-analysis would not have been feasible or would have required excessive narrowing of scope. The narrative review allows also a broader interpretative perspective that we consider essential for the practical clinical implications. Limitations of this method have been discussed, and stressed, in the last section of the manuscript, as reported in the statement “Some limitations need to be acknowledged. As a narrative review, this study lacks the methodological rigor of a systematic review and is therefore potentially subject to selection bias and other inherent limitations related to the non-systematic inclusion of studies”. For these reasons, we chose to include the term “narrative” in the title to accurately reflect the methodological approach of this review, and we believe that specifying its narrative nature is important for clarity and transparency. In the Methods section we also justified the choice of this approach: A narrative review format was adopted to provide a broad overview of the heterogeneous literature on prognostic factors in MS, integrating evidence across various clinical, molecular, and radiological domains. Due to the wide variability in study designs, outcomes, and cohorts, the narrative approach allowed a wider interpretative perspective, essential to contextualize the clinical implications of this review.

Comment 3. Key concepts like “PIRA,” “RAW,” and “smoldering MS” are introduced with some explanation in introduction section but lack precise definitions or clinical thresholds. These should be more clearly delineated for clarity and scientific rigor.

We thank the Reviewer for this thoughtful observation. We fully agree on the crucial role of smouldering disease mechanisms and the contribution of PIRA to long-term disability accrual, starting from the earliest phases of MS. These aspects have been addressed across several sections of the review, including: the discussion of radiological biomarkers of smouldering MS, the role of biomarkers in capturing PIRA-related processes, and the impact of DMTs on both relapse-dependent and relapse-independent disability progression. In the introduction, we also referred to a recent publication from our group, published in this Journal, which provides an in-depth and up-to-dated discussion on these topics: Guerra, T.; Iaffaldano, P. A Window into New Insights on Progression Independent of Relapse Activity in Multiple Sclerosis: Role of Therapies and Current Perspective. Int J Mol Sci. 2025, 26(3):884. https://doi.org/10.3390/ijms26030884. Therefore, to avoid redundancy, we did not further elaborate on the individual definitions in this manuscript but rather presented them in the context of their prognostic relevance.

Comment 4. Although the final sentences of introduction hint at the review’s purpose, the aim is still unclear. The authors should explicitly state the key questions being addressed and clarify the scope to sharpen the reader’s expectations.

We thank the Reviewer for this valuable suggestion, which allowed us to clarify and better articulate the objectives of our study. Therefore, we have revised the final part of the introduction section, highlighting our aim to examine and summarize the current evidence regarding the most robust clinical, molecular, and radiological prognostic factors in adult-onset MS, and to discuss their relevance in light of clinical practice. The revised paragraph reads as follows: “The purpose of this study is to present an extensive and updated narrative review of evidence related to identified prognostic factors for adult-onset MS. The aim is to gather and thematically organize prognostic factors in MS, providing a synthesis that is relevant both from a clinical and research perspective”. We believe that this sentences can be also linked to the specifications added to the first part of the Methods section.

Comment 5. Time frame of collected studies should be mentioned in 2.2 section.

We acknowledge the Reviewer's observation. We have now revised the Methods sections specifing the time frame, with the following revision: Specifying the time frame of the search, no restrictions were placed on publication dates, but we focused mainly on studies published in the last years. The time span of the references included in this review ranges from 2003 to 2025. Notably, the majority of the cited literature (approximately 70%) was published from 2020 onward, reflecting the most recent advancements in the field.

Comment 6. Several terms in the manuscript are introduced in their abbreviated form without first being defined. For clarity and to ensure accessibility for a broad scientific readership, all abbreviations should be clearly defined at their first mention.

We acknowledge the Reviewer’s observation. We have corrected these incongruencies throughout the text and we have also included a list of abbreviations in the Manuscript, at the end of the text and before references.

Comment 7. Grammatical refinements are needed throughout the manuscript to enhance clarity and readability.

We thank the Reviewer for the scrutiny of our paper. We have thoroughly revised the manuscript from a linguistic and grammatical perspective, improving sentence structure with grammatical refinements. The revisions are highlighted in yellow in the tracked changes documents and included in the text of the revised file.

Reviewer 4 Report

Comments and Suggestions for Authors

Dear Editor,

Thank you for granting me the opportunity to review this narrative review, which aims to identify prognostic factors for adult-onset MS, thereby highlighting unmet needs and challenges in the management of MS treatment. While the manuscript addresses an important topic, several issues need to be addressed before it can be considered for publication:

  1. Formatting Issues in Title Page:
    There are special colors (yellow and green) highlighting some authors on the title page. Does this indicate these areas require revision? The authors should avoid such confusing formatting errors before resubmission.

  2. Study Design Clarity:
    In section 2.2 (Study Design), although the authors provide detailed references for the studies included, a flowchart is needed to enhance the readability and transparency of this section.

  3. Exclusion of Environmental Factors:
    The authors state that environmental exposures were not considered in this analysis. Given the growing body of evidence suggesting that modifiable lifestyle and environmental factors may have a quantifiable impact on MS prognosis, could the authors provide a reasonable justification for not investigating these factors?

  4. Ethnic Background Not Addressed:
    In the section on demographic and clinical prognostic factors, ethnic background was neglected. Are there differences in MS prognosis among minority ethnic groups, particularly Black and Latino/Hispanic individuals, compared to other groups? This should be discussed.

  5. Radiological Predictors:
    T1-hypointense lesions ("black holes"), an important radiological sign in MS, were not discussed. Additionally, I agree with the authors that lesion topography is an important prognostic factor. The authors could cite MRI features from published papers as a supplementary file to help readers better understand these radiological predictors.

  6. Cognitive Impairment:
    Cognitive impairment (CI) is a common and clinically significant manifestation of adult-onset MS, and should also be discussed in this review.

  7. Practical Recommendations and Implementation:
    A section discussing the practical recommendations and implementation of the identified prognostic factors is needed. For example, the Barcelona Baseline Risk Score (BRS) is a recent, externally validated tool that uses flexible combinations of predictors—including age at first attack, sex, number and topography of T2 lesions, and disability at first visit—to estimate time to EDSS 3.0 and stratify patients into risk groups with significantly different long-term outcomes. Including such tools would highlight the clinical value of this study.

Overall, the manuscript addresses a significant topic, but the above issues should be carefully addressed to improve its clarity, completeness, and clinical relevance.

Best regards,

Author Response

Response to Reviewer 4 Comments

The Reviewer's time spent reviewing this manuscript is greatly appreciated. Please find the detailed responses below and the corresponding revisions highlighted in track changes in the re-submitted files.

Comment 1. Formatting Issues in Title Page: There are special colors (yellow and green) highlighting some authors on the title page. Does this indicate these areas require revision? The authors should avoid such confusing formatting errors before resubmission.

We thank the Reviewer for the comment. We have not included highlighted texts in the original submitted file in Authors sections, so the different colors are due to a post-editorial check after submission sent to peer-review. We have removed the colors in the revised Manuscript file.

Comment 2. Study Design Clarity: In section 2.2 (Study Design), although the authors provide detailed references for the studies included, a flowchart is needed to enhance the readability and transparency of this section.

We appreciate and acknowledge the reviewer's observation, allowing us to include this additional figure in our review. Therefore, we created a figure (Figure 1) describing the process of article selection with a flowchart.

Comment 3. Exclusion of Environmental Factors: The authors state that environmental exposures were not considered in this analysis. Given the growing body of evidence suggesting that modifiable lifestyle and environmental factors may have a quantifiable impact on MS prognosis, could the authors provide a reasonable justification for not investigating these factors?

We thank the reviewer for this comment which enable us to better clarify this topic.  We fully agree with the Reviewer that modifiable lifestyle and environmental factors may have a significant prognostic impact in multiple sclerosis. However, as specified in the Funding section, this work is part of a broader project aimed at investigating prognostic factors in MS: the PROMISING Study. Within the framework of this project, additional related papers, currently under development, will specifically address environmental and genetic factors. Given the relevance of the Reviewer’s comment, we have expanded the statement in the Methods section (in the last paragraph of funding information) to clarify the rationale behind the exclusion of these variables from the scope of the present discussion.

Comment 4. Ethnic Background Not Addressed: In the section on demographic and clinical prognostic factors, ethnic background was neglected. Are there differences in MS prognosis among minority ethnic groups, particularly Black and Latino/Hispanic individuals, compared to other groups? This should be discussed.

We thank the reviewer for this suggestion, useful to expand, in our manuscript, the discussion about the challenging topic of prognosis in MS. In the section of demographic ad clinical prognostic factors, we included the following text: “People of all ethnicities are affected by MS, and the ethnic background must be included in the discussion about prognosis. A recent review outlined that Black, Latino/Hispanic, and South Asian individuals with MS in North America and the UK appear to have an earlier age of onset. Furthermore, compared to white MS patients, Black and Latino/Hispanic MS patients in the USA are more likely to have severe symptoms at disease onset and an earlier disability accrual [Li, J., Vas, N., Amezcua, L. et al. Multiple Sclerosis in People of Diverse Racial and Ethnic Backgrounds: Presentation, Disease Course, and Interactions with Disease-Modifying Therapy. CNS Drugs (2025)]. Notable differences in MS-specific mortality trends by age and race/ethnicity were also highlighted, indicating an unequal burden of disease and a complex balance of environmental and social differences influencing disease variability [Amezcua L, McCauley JL. Race and ethnicity on MS presentation and disease course. Multiple Sclerosis Journal. 2020;26(5):561-567]”

Comment 5. Radiological Predictors: T1-hypointense lesions ("black holes"), an important radiological sign in MS, were not discussed. Additionally, I agree with the authors that lesion topography is an important prognostic factor. The authors could cite MRI features from published papers as a supplementary file to help readers better understand these radiological predictors.

We sincerely thank the reviewer for the scrutiny of our review, allowing us to better discuss this topic. Considering the triggering topic of SEL, we included the following sentence: “In a recent study, the proportion of persisting black holes was higher in SELs compared to non-SELs, and within-patient SEL and persisting black holes volumes were positively correlated [Calvi A, Tur C, Chard D, et al. Slowly expanding lesions relate to persisting black-holes and clinical outcomes in relapse-onset multiple sclerosis. Neuroimage Clin. 2022;35:103048. doi:10.1016/j.nicl.2022.103048]". We have also included the following sentence, which gathers multiple sources from literature, in consideration of the predictive role of T1-hypointense lesions: “T1-hypointense lesions primarily indicate axonal degeneration, white matter disruption, and are typically linked to irreversible clinical outcomes [Bagnato F, et al. Evolution of T1 black holes in patients with multiple sclerosis imaged monthly for 4 years. Brain. 2003; 126(Pt 8): 1782-1789; Sahraian M, et al. Black holes in multiple sclerosis: definition, evolution, and clinical correlations. Acta Neurol Scand. 2010; 122(1): 1-8; Valizadeh A, et al. Correlation between the clinical disability and T1 hypointense lesions' volume in cerebral magnetic resonance imaging of multiple sclerosis patients: A systematic review and meta-analysis. CNS Neurosci Ther. 2021;27(11):1268-1280]”. We also appreciate the Reviewer’s thoughtful suggestion regarding the inclusion of relevant references on radiological prognostic factors. To avoid further lengthening Table 1, and to avoid excessive polarization around the MRI topic, in light of the review’s integrated perspective, we refined and clearly labeled the entries related to neuroimaging biomarkers (First column, “Prognostic factor”)..

Comment 5. Cognitive Impairment: Cognitive impairment (CI) is a common and clinically significant manifestation of adult-onset MS and should also be discussed in this review.

We appreciate and acknowledge the Reviewer's observation. In the first submitted version of the manuscript, we had already included some elements regarding the prognostic relevance of cognitive decline, with a specific focus on this aspect. In the revised version, we have expanded this section by adding further references and integrating the discussion more comprehensively into the overall framework of prognostic factors, as follow: “This discussion ought to cover cognitive impairment as predictors of the worst out-comes. Chronic depression and cognitive dysfunction were associated with adverse long-term outcomes in MS [37]. Cognitive impairment was associated with higher odds of transitioning from relapsing–remitting course to a progressive disease course, and interestingly, it was also associated with a higher mortality risk [43]. An Italian 10-year retrospective longitudinal study outlined that cognitive impairment at diagnosis, with particular involvement of memory and processing speed, was associated with a more than threefold risk of reaching EDSS 4.0 and a twofold risk of SPMS conversion [Moccia, M.; Lanzillo, R.; Palladino, R.; Chang, K. C.; Costabile, T.; Russo, C.; De Rosa, A.; Carotenuto, A.; Saccà, F.; Maniscalco, G. T.; et al. Cognitive impairment at diagnosis predicts 10-year multiple sclerosis progression. Mult Scler. 2016, 22(5), 659–667]. Cognitive impairment can occur independently of other neurological symptoms and is linked to a higher risk of future neurological disability [Benedict, R. H. B.; Amato, M. P.; DeLuca, J.; Geurts, J. J. G. Cognitive impairment in multiple sclerosis: clinical management, MRI, and therapeutic avenues. The Lancet. Neurology 2020, 19(10), 860–871].

Comment 5. Practical Recommendations and Implementation: A section discussing the practical recommendations and implementation of the identified prognostic factors is needed. For example, the Barcelona Baseline Risk Score (BRS) is a recent, externally validated tool that uses flexible combinations of predictors—including age at first attack, sex, number and topography of T2 lesions, and disability at first visit—to estimate time to EDSS 3.0 and stratify patients into risk groups with significantly different long-term outcomes. Including such tools would highlight the clinical value of this study.

We thank the reviewer for these wise suggestions. We thank the Reviewer for the helpful comment. We have renamed Section 8 to better reflect the focus of the discussion on prognostic tools: “The Strategic Role of Prognostic Algorithms in Clinical Decision-Making and Research”. This change aligns with the aim of the section, which was specifically intended to address prognostic tools. Accordingly, we have also incorporated the risk score suggested by the Reviewer into the discussion. The following sentences are now included in the text: “A very recent prognostic tool is the Barcelona Risk Score (BRS), a validated algorithm that incorporates several biomarkers to classify each patient in four data-driven groups according to the risk of moderate long-term disability, considering different outcomes: RAW, PIRA, SPMS conversion, MRI features and patient-reported scores. The score offers a versatile framework, designed to support clinical decision-making in everyday practice and across heterogeneous settings, and it’s applicable even with limited data availability. [The Barcelona baseline risk score to predict long-term prognosis after a first demyelinating event: a prospective observational study. Tur, Carmen et al. The Lancet Regional Health – Europe, Volume 53, 101302]. Following the Reviewer suggestion, we stressed also the concept of a practical recommendation, in light of the discussion about prognosis throughout the text of our review: In conclusion, our practical recommendation is to adopt integrative algorithms capable of assessing the patient comprehensively, by combining biomarkers that reflect the disease from multiple angles, ranging from clinical presentation to molecular indicators of progression, and neuroimaging findings.

Finally, we have also thoroughly revised the manuscript from a linguistic and grammatical perspective, improving sentence structure and correcting minor typographical errors.

Round 2

Reviewer 3 Report

Comments and Suggestions for Authors

no

Reviewer 4 Report

Comments and Suggestions for Authors

I have no more question at this stage